# $E^3$: Exploring Embodied Emotion Through A Large-Scale Egocentric Video Dataset

**Wang Lin**[1,*], **Yueying Feng**[1,*], **Wenkang Han**[1,*], **Tao Jin**[1], **Zhou Zhao**[1],
**Fei Wu**[1], **Chang Yao**[1], **Jingyuan Chen**[1,†],
Zhejiang University[1]
{linwanglw, yueyingf, jint_zju, zhaozhou,
wufei, changy, jingyuanchen,}@zju.edu.cn

## Abstract

Understanding human emotions is fundamental to enhancing human-computer interaction, especially for embodied agents that mimic human behavior. Traditional emotion analysis often takes a third-person perspective, limiting the ability of agents to interact naturally and empathetically. To address this gap, this paper presents $E^3$ for Exploring Embodied Emotion, the first massive first-person view video dataset. $E^3$ contains more than 50 hours of video, capturing 8 different emotion types in diverse scenarios and languages. The dataset features videos recorded by individuals in their daily lives, capturing a wide range of real-world emotions conveyed through visual, acoustic, and textual modalities. By leveraging this dataset, we define 4 core benchmark tasks - emotion recognition, emotion classification, emotion localization, and emotion reasoning - supported by more than 80k manually crafted annotations, providing a comprehensive resource for training and evaluating emotion analysis models. We further present Emotion-LlaMa, which complements visual modality with acoustic modality to enhance the understanding of emotion in first-person videos. The results of comparison experiments with a large number of baselines demonstrate the superiority of Emotion-LlaMa and set a new benchmark for embodied emotion analysis. We expect that $E^3$ can promote advances in multimodal understanding, robotics, and augmented reality, and provide a solid foundation for the development of more empathetic and context-aware embodied agents. Project page: https://exploring-embodied-emotion-official.github.io.

## 1  Introduction

Embodied Agents [16], which are capable of simulating human interactions by perceiving the environment and executing corresponding behaviors, have facilitated a wide range of applications [53, 55, 56, 58]. Emotion analysis [39], as a key component of understanding human behavior, is crucial for improving the naturalness and effectiveness of human-computer interaction. To enhance the accuracy of emotional interactions in real-world scenarios, embodied agents necessitate large and diverse datasets for training and validating their algorithms.

Although current emotion analysis techniques have made significant progress in a variety of applications, most works [90, 98, 105] focus on analyzing emotions from a *third-person view*, which involves assessing emotional states from an external observer's standpoint. However, for embodied agents, understanding emotions from a *first-person view* (FPV) or an *egocentric* perspective is crucial [60]. In FPV, agents perceive and respond to emotions through their own perceptual system, which is more

---

[*] Equal Contribution.
[†] Corresponding Author.

38th Conference on Neural Information Processing Systems (NeurIPS 2024) Track on Datasets and Benchmarks.

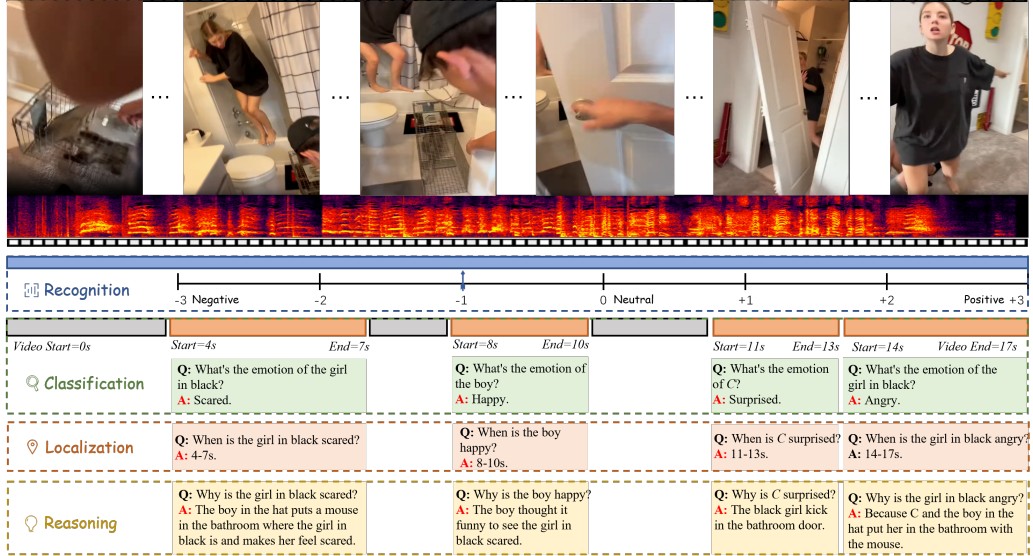

Figure 1: **Examples in** $E^3$. $E^3$ collects massive first-person view videos in which the camera wearer (denoted as $C$) actively engages in the video activities. Each video in the dataset is annotated manually with fine-grained labels to support four embodied emotion analysis tasks.

aligned with human experience. Analyzing emotions from an FPV can provide agents with more personalized and direct emotional information, enabling them to generate more accurate responses.

The goal of this work is to advance research on emotion analysis from an FPV and provide solid support for the emotion interaction capabilities of embodied agents. Our primary contribution is the creation of the dataset $E^3$: a large-scale collection of FPV videos that capture people's activities and emotions in daily life, as shown in Figure 1. The dataset contains 50 hours of videos covering 8 emotion types in multiple languages including Mandarin, Cantonese, and English. The camera wearers include people of different genders, ages, and occupations, while the video backgrounds cover diverse daily scenes such as home, office, city, and countryside. Meanwhile, the dataset also includes unedited videos and post-edited videos, aiming to support a wider and more robust scenario for emotion analysis. In addition to RGB videos, all data samples are equipped with full multimodal information, including video titles, audio, audio transcribed text, etc., allowing combining multiple modalities together for emotion analysis. As shown in Table 1, $E^3$ addresses a significant deficiency in emotion analysis in egocentric videos by enhancing scale, detail, diversity, and robustness.

Our second contribution involves the introduction of four benchmark tasks as shown in Figure 1 that constitute the core components of embodied emotion analysis - including:

- **Emotion Recognition**: determining the overall emotion tendency of the video.
- **Emotion Classification**: identifying the specific emotion category of each person in the video, including the camera wearer.
- **Emotion Localization**: locating the start and end times of emotion based on a given textual query like *"when does the girl in black feel happy"*.
- **Emotion Reasoning**: reasoning the cause of the emotion in the context based on a given textual query like *"why is the girl in black afraid"*.

To facilitate these tasks, the dataset is equipped with over 20k annotations derived from extensive manual labor totaling more than 4,000 hours. These annotations cover semantic and temporal labels, as well as emotion strength, query texts, and human explanations for emotions.

Our third contribution is the construction of a video understanding framework, named **Emotion-LlaMa**, designed for embodied emotion analysis. FPV video analysis requires a comprehensive understanding of the camera wearer's physical environment and the ability to interpret emotions within a human context. Given that the camera wearer is often not visible within the frame, it is essential to integrate multiple modalities in order to fully understand the video content. However, existing video understanding models [7, 32, 45, 108] are typically trained on third-person view

Table 1: Comparison of our proposed $E^3$ dataset with mainstream emotion analysis datasets (top) and egocentric datasets (bottom).

| Dataset | Domain | Dur(hrs) | #labels | Modality | Language | Emotion? | Ego? | Example |
|---|---|---|---|---|---|---|---|---|
| EmoDB[89] | speech | 03:00 | 800 | a | DE | ✓ | ✗ | |
| Large Movie[51] | movie | - | 25,000 | t | EN | ✗ | ✗ | |
| SeMAINE[54] | dialogue | 06:30 | 80 | v,a | EN | ✓ | ✗ | |
| HUMAINE[84] | diverse | 04:11 | 50 | v,a | various | ✓ | ✗ | |
| YouTube[57] | diverse | 00:29 | 300 | v,a,t | various | ✗ | ✗ | Emotion in 3rd-person view |
| SST[80] | movie | - | 11,855 | t | EN | ✗ | ✗ | |
| ICT-MMMO[96] | movie | 13:58 | 340 | v,a,t | EN | ✗ | ✗ | |
| RECOLA[76] | dialogue | 03:50 | 46 | v,a | FR | ✓ | ✗ | |
| MOUD[65] | review | 00:59 | 400 | v,a,t | ES | ✗ | ✗ | |
| AFEW[14] | movie | 02:28 | 1,645 | v,a | various | ✓ | ✗ | |
| MOSI[101] | diverse | 02:36 | 2,199 | v,a,t | EN | ✓ | ✗ | No emotion in 1st-person view |
| MOSEI[102] | diverse | 65:53 | 23,453 | v,a,t | EN | ✓ | ✗ | |
| SEWA[38] | adverts | 04:39 | 538 | v,a | EN,DE,EL | ✓ | ✗ | |
| CH-SIMS[99] | adverts | - | 2281 | v,a,t | CH | ✓ | ✗ | |
| Disneyworld[18] | disneyland | 42:00 | 15,000 | v,a,t | EN | ✗ | ✓ | |
| EGTEA Gaze+[19] | diverse | 28:00 | - | v,a,t | various | ✗ | ✓ | |
| BEOID[8] | diverse | - | - | v,a,t | EN | ✗ | ✓ | |
| Charades-Ego[78] | home | 34:00 | 30,000 | v,a,t | EN | ✗ | ✓ | |
| EPIC[10] | kitchen | 100:00 | 90,000 | v,a,t | EN | ✗ | ✓ | |
| Ego-4D[22] | diverse | 3025:00 | 74000 | v,a,t | various | ✗ | ✓ | Emotion in 1st-person view |
| $E^3$(Ours) | **diverse** | **71:41** | **81,248** | **v,a,t** | **various** | ✓ | ✓ | |

datasets and are primarily designed for activity recognition, emphasizing visual cues while ignoring the acoustic modality. To bridge this gap, our Emotion-LlaMa incorporates an audio branch that captures information such as tone and rhythm, which can be instrumental in emotion analysis, particularly in discerning the emotions of the camera wearer.

We fine-tune Emotion-LlaMa and existing models on $E^3$. Experiments demonstrate the superiority of Emotion-LlaMa and sets the first benchmark for embodied emotion analysis. We expect $E^3$ can pave the way for building embodied agents capable of seamlessly learning human emotions. These agents will be empowered with empathy and seamlessly integrate into human daily interactions.

## 2 Related Work

**Egocentric Video Datasets.** Over the past decade, multimodal understanding [33, 48, 49, 64] has received much attention, numerous egocentric datasets [9, 11, 41, 67] have been developed, presenting a variety of challenging research topics [6, 11, 20, 35, 41]. However, due to the high cost associated with collecting egocentric videos, previous datasets have typically been small-scale and focused on specific domains. Several prominent egocentric datasets [12, 19, 46, 67] have demonstrated the potential of first-person views for action recognition, particularly in hand-object interactions. Conversely, larger egocentric datasets have primarily focused on kitchen environments [13, 44]. Recently, the release of a substantial egocentric video dataset, Ego4D [22], comprising 3,670 hours of video content across various indoor and outdoor settings, has attracted attention. However, these videos often lack emphasis on human dialogue and facial expressions, focusing more on actions. This lack of interpersonal communication and emotion expression makes them unsuitable for emotion analysis. In contrast, $E^3$ features diverse scenes with rich human interactions and emotion expressions, bridging the gap in FPV emotional datasets.

**Emotional Analysis Datasets.** Emotion analysis datasets serve as a crucial foundation for advancing research in emotion computing and human-computer interaction. These datasets [37, 68, 97, 101] are commonly sourced from movies, customer reviews, video, and speech video. One of the pioneers in this field is the IEMOCAP [4] dataset, offering a comprehensive multimodal collection for emotion analysis. The DEAP [37] dataset builds upon this by incorporating EEG signals, enabling the study of physiological reactions to emotional stimuli. The MOSI [101] dataset and its successors, such as CMU-MOSEI [102], have further contributed by sourcing data from YouTube videos, annotated with emotional intensity scores, thus facilitating the analysis of emotional expressions in natural settings. The introduction of cross-lingual datasets like CMU-MOSEAS [100], which includes languages such as Spanish(ES), Portuguese(PT), German(DE), and French(FR), has also expanded the scope of emotion analysis research to better accommodate a diverse and global user base.

Despite these advancements, current datasets present limitations, particularly for applications in robotics and augmented reality where inputs are in first-person view. While existing datasets typically feature third-person perspectives, capturing activities without the camera wearer's involvement, thereby impeding the development of embodied agent empathy. Furthermore, exiting datasets focus on emotion recognition and categorization, neglecting the moments and causes of emotions. This oversight results in a superficial understanding of emotion in existing models, indicating a need for datasets that capture first-person experiences and provide deeper insights into the context of emotions.

## 3 $E^3$ for Exploring Embodied Emotion

The dataset, denoted as $E^3$, serves as a comprehensive resource for multiple emotion analysis tasks. It comprises a collection of $21,998$ videos, each ranging from 10 to 30 seconds in duration, accompanied by human annotations for emotion analysis. This section delves into the details of our data collection and annotation procedures.

### 3.1 Data Collection and Preprocessing

Videos that satisfy embodied emotion analysis should have both 1) a first-person perspective recording the social activities of the videomaker, and 2) a clear expression of emotion. The videos are collected from 3 popular websites: `YouTube`, `BiliBili` and `Douyin`, with a focus on video blogs (vlogs) that document the daily lives of the individuals behind the camera. These vlogs are typically tagged with the hashtag *#vlog*. The videos in our dataset were recorded in a variety of settings and with different equipment. Some users utilized high-quality microphones and cameras, while others opted for less professional recording devices. The distance between the camera and the user varied, as did the background and lighting conditions. All videos are maintained in their original resolution and saved in MP4 format. The length of the videos ranges from 3 to 5 minutes.

To facilitate the annotation process, we segment the videos into shorter clips through shot detection, typically ranging from 10 to 30 seconds. These clips encapsulate the entire event and serve as the fundamental unit for analysis. Note that the segmentation is seamless, allowing researchers to easily reassemble the clips for long-term video analysis. Moreover, in cases where the videos do not contain subtitles, we employ Whisper [73] for speech recognition. This technology enables us to generate subtitles for the videos, aiding annotators in their labeling process.

### 3.2 Annotation Construction

We utilize a third-party vendor to annotate our data (see more details in Appendix). Prior to the annotation process, human annotators go through a strict training phase. We employed a total of 26 people, including 14 females and 12 males, and a variety of educational backgrounds to cover most human understanding of emotions. The entire annotation process took more than $5,500$ hours.

**Stage 1: Emotion Recognition Annotation.** In this stage, the annotator first filters videos that do not meet the embodied emotion analysis, such as videos that are not in FPV or where the camera wearer is not involved in the activity. Then they are asked to determine whether the emotion of the whole video is positive or negative with an emotion intensity score [101], which is defined from strongly negative to strongly positive with a linear scale from $-3$ to $+3$. Considering that the judgment of emotion intensity is very subjective, we employ 3 annotators to annotate the same video, and their average scores are rounded as the final annotation. Finally, videos with an emotion intensity of $0$, *i.e.,* videos with no recognizable emotion, are also excluded from further analysis.

**Stage 2: Emotion Classification Annotation.** Following the completion of Stage 1, the annotator proceeds to further annotate the emotions exhibited by individuals in the video. A total of 8 emotion categories were designed, with markers denoting emotion intensity ranging from weak to strong (1 to 3). Six of these emotion categories, namely *{Happy, Sad, Angry, Scared, Disgusted, and Surprised}* are from Ekman [17]. The other two emotions *{Shy, and Sarcastic}* are identified as hidden emotions based on feedback from annotators. Hidden emotions pose a significant challenge in emotion analysis tasks, as they require contextual analysis for accurate interpretation. To the best of our knowledge, our dataset is the first to include hidden emotions in emotion analysis.

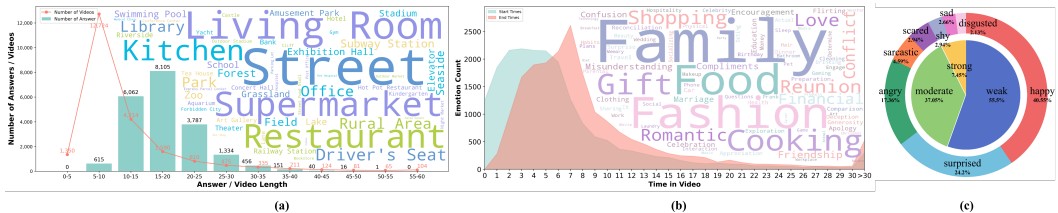

Figure 2: Visualization of static analysis of $E^3$: **(a)** Distribution of video duration, answer length and scenes; **(b)** Distribution of emotions timestamps and topics; **(c)** Distribution of emotion categories.

Each video typically involves multiple characters, including the camera wearer and other individuals who appear in the video. To facilitate emotion analysis of these characters, we opt to distinguish them based on their physical attributes, such as "woman with glasses", rather than using specific identity nouns like *mother* or *boyfriend*.

**Stage 3: Emotion Localization Annotation.** In this stage, the annotators identify the specific moments when emotions begin and end for each character based on the categorization of emotions assigned to each character in Stage 2. The start and end times of these emotions are pinpointed by paying attention to the corresponding sounds or expressions made by the characters. It is important to note that emotions from multiple characters may overlap during the same time period, as they are actively engaged in a real-time socialization activity.

**Stage 4: Emotion Reasoning Annotation.** In this stage, the annotator is tasked with analyzing the specific reasons behind each emotion exhibited by the characters, based on the contextual information provided. To formulate the question text, a standardized template of "*why [person] [emotion]?*" is used. This helps guide the annotator in identifying the underlying reasons for the character's emotions. In the answer text, the annotator is required to objectively describe the event that triggered the emotion and explain the relationship between this event and the character's emotional response. To ensure the accuracy and quality of the answer text, a separate group of 10 annotators will review it for spelling mistakes and grammatical errors.

### 3.3 Dataset Analysis

**Statistics.** $E^3$ is the first egocentric video emotion analysis dataset. Table 1 illustrates a comparison between $E^3$ and other existing video datasets. $E^3$ introduces a unique perspective by providing first-person video data, where the camera wearer actively participates in emotion-related activities rather than observing from a distance. This dataset offers a more direct view of the facial expressions and body movements of individuals involved in interactions, thereby enhancing the training of empathetic embodied agents. In terms of annotated labels, $E^3$ contains 4 tasks with a total of 80k annotations, making it the largest video dataset for emotion analysis. Furthermore, $E^3$ features diverse linguistic content, including *Mandarin*, *English*, as well as languages such as *Cantonese*, *Japanese*, and *Korean* that are not present in other datasets.

Compared to existing FPV video datasets, $E^3$ bridges a crucial gap in the field of emotion analysis for embodied agents. While both datasets offer comprehensive multimodal information, $E^3$ focuses more on social interactions, whereas the first-person video dataset focuses more on actions. As a result, the acoustic (tone of voice, emotional expressions like *crying* and *laughing*) and textual (dialogues) modalities in $E^3$ contain richer information.

**Distribution of Videos and Answer Texts.** In total, we have annotated 18k videos and over 70k labels. Figure 2(a) shows the distribution of video durations and answer lengths. The average duration of videos is 11.5s and the average length of answer texts is 18 words. Notably, the average video duration of $E^3$ is longer compared to existing emotion analysis datasets, which typically have an average video duration of 6.2s. This indicates that our videos contain more emotional context information. Additionally, Figure 2(a) illustrates the distribution of video scenes, with $E^3$ covering a wide range of daily scenes such as living rooms, streets, and offices.

**Distribution of Emotions.** Figure 2(b) shows the distribution of emotion timestamps and video topics. Most emotions happen within the 2nd-5th seconds of the video, and the average duration of emotions is 2.62s. Specifically, the shortest duration is observed for the emotion *surprised* at 1.68s,

while emotions like *happy* and *sarcastic* tend to last longer, averaging 2.71s and 3.39s, respectively. Moreover, the video content of $E^3$ encompasses a variety of topics including *family*, *food*, and *shopping*. Figure 2(c) displays the distribution of the 8 emotions in the video clips of $E^3$. On average, each video clip contains 1.32 emotions. The emotion *happy* is the most frequently occurring emotion (40.55%), while *disgusted* is the least prevalent emotion (2.13%). Additionally, each emotion was assigned an intensity level ranging from 1 to 3, with the majority of emotions exhibiting an intensity of 1, totaling at $10,953$.

### 3.4 Privacy Protection

When constructing $E^3$, we follow the principles of privacy protection and copyright respect. We obtain explicit consent from video owners and ensure that they understand the use of personal information (*e.g.*, portrait, voice). We comply with the data protection regulations of *YouTube*, *Douyin*, and *BiliBili*, providing transparency to participants and guaranteeing their right to withdraw consent. We collect only necessary data and ensure strict data usage boundaries to support research while protecting privacy. Further details can be found in Appendix.

## 4 Methods

We first briefly review the structure of existing multimodal Large Language models (MLLMs) for video understanding. Subsequently, we introduce **Emotion-LlaMa**, a novel model integrates the acoustic modality to enhance the emotion comprehension of embodied videos.

**Preliminary.** MiniGPT4-Video [1] is a significant advancement in the field of MLLMs tailored for video understanding. Building upon the foundation established by LlaMa [86], MiniGPT4-Video utilizes EVA-CLIP [83] to align visual modality with textual modality and maps them to the language space in LlaMa using a linear layer. Notably, MiniGPT4-Video optimizes efficiency by consolidating 4 adjacent visual tokens in each frame into a single token, thereby reducing the overall number of tokens. By combining visual token

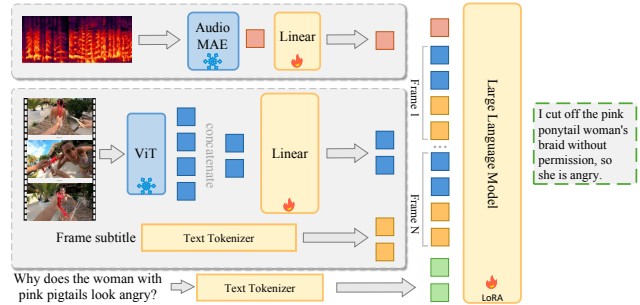

Figure 3: **Overview of Emotion-LlaMa.**

of each sampled frame with the corresponding textual token from frame subtitle, MiniGPT4-Video enables LlaMa to better interpret video content.

**Emotion-LlaMa.** In this study, we propose Emotion-LlaMa, an extended model based on MiniGPT4-Video, aiming to analyze emotions in an FPV video. It is important to note that relying solely on visual modality for embodied emotion analysis may be insufficient for several reasons. First, the camera wearers are often not visible in the frame, which limits the visual cues available for emotion inference. Second, there exists a strong correlation between audio cues such as pitch and rhythm and emotional expression. Therefore, it is essential to incorporate acoustic modality into the emotion analysis process. To address this, we have incorporated an audio feature branch into MiniGPT4-Video. Specifically, we employ an advanced audio encoder, AudioMAE [30], to convert audio inputs to audio features. Then these audio features are mapped to the language space of LLMs through an audio linear layer. In this way, Emotion-LlaMa can combine multimodal information to comprehensively analyze emotions in FPV videos.

## 5 Benchmarks

In this section, we present a challenging suite of benchmark tasks. The four benchmark tasks include 1) embodied emotion recognition; 2) embodied emotion categorization; 3) embodied emotion localization; and 4) embodied emotion reasoning. Each task and the specialized baselines are

Table 2: Results of **embodied emotion recognition** task.

| Methods | Acc | F1 |
|---------|-------|-------|
| MMIM | 73.12 | 75.51 |
| CENET | 75.60 | 77.17 |
| ALMT | 77.11 | 78.72 |
| MALMM | 47.82 | 61.33 |
| MALMM | 53.50 | 67.02 |
| VAST | 81.50 | 91.33 |
| VAST | 84.92 | 92.71 |
| MINI | 80.91 | 90.52 |
| MINI | 83.83 | 92.64 |
| Ours | **85.93** | **93.70** |

Table 3: Comparing different approaches on **embodied emotion classification** task. Fine-tuning methods are highlighted in grey.

| Methods | happy | angry | surprised | shy | sarcastic | sad | disgusted | scared | avg |
|---------|-------|-------|-----------|------|-----------|-------|-----------|--------|-------|
| DFAN | 44.63 | 37.26 | 35.79 | 9.80 | 6.42 | 11.57 | 10.17 | 3.32 | 37.76 |
| VAANet | 56.85 | 40.52 | 32.83 | 5.26 | 7.01 | 3.77 | 7.01 | 6.74 | 43.30 |
| CTEN | 52.72 | 40.14 | **40.10** | 6.59 | 9.41 | 6.54 | 5.56 | 10.20 | 42.37 |
| MALMM | 57.37 | 1.72 | 14.55 | 8.00 | 2.38 | 7.29 | 0 | 0 | 35.44 |
| MALMM | 3.58 | 16.48 | 31.81 | 5.48 | 5.33 | 6.06 | **12.50** | 2.53 | 18.10 |
| VAST | 53.51 | 5.33 | 1.68 | 5.34 | 0 | 16.67 | 0 | 3.85 | 33.51 |
| VAST | 63.27 | 40.15 | 3.88 | 8.96 | 8.53 | 3.39 | 11.76 | 11.24 | 43.55 |
| MINI | 54.29 | 5.25 | 1.23 | 5.03 | 0 | 21.05 | 0 | 2.78 | 35.41 |
| MINI | 61.88 | **43.40** | 0.62 | **15.62** | 4.65 | 6.78 | 17.14 | 4.71 | 44.95 |
| Ours | **64.45** | 41.48 | 16.94 | 9.84 | **9.62** | 6.78 | 11.76 | **11.36** | **45.62** |

Table 4: Comparing different approaches on **embodied emotion localization** task. '-' indicates that LLMs based models cannot generate multiple localization candidates simultaneously.

| Methods | R1,U@0.3 | R1,U@0.5 | R1,U@0.7 | R5,U@0.3 | R5,U@0.5 | R5,U@0.7 | mIoU |
|---------|----------|----------|----------|----------|----------|----------|-------|
| MSAT[104] | 31.74 | 15.69 | 7.11 | 66.05 | 39.71 | 19.73 | 25.41 |
| r2-tuning[50] | 33.62 | 17.18 | 7.61 | 69.57 | 38.53 | 19.26 | 21.73 |
| VTG-LLM[25] | 15.46 | 7.73 | 1.59 | - | - | - | 12.01 |
| VTG-LLM[25] | 28.68 | 17.61 | 5.14 | - | - | - | 19.42 |
| TimeChat[75] | 18.28 | 6.87 | 1.84 | - | - | - | 14.94 |
| TimeChat[75] | 30.31 | 17.67 | 5.76 | - | - | - | 20.58 |

described below. Additionally, detailed implementation instructions and qualitative examples are included in Appendix.

## 5.1 Embodied Emotion Recognition

**Motivation.** The ability to recognize emotions is crucial for embodied agents, as it enables them to understand the emotional tone of interactions. By being able to perceive the emotional tone of ongoing activities, these agents can provide responses that are sensitive to the emotional context, showing empathy and enhancing the user experience. This capability is essential for creating emotionally intelligent interactions, which in turn improves the relationship between the user and the agent.

**Task Definition.** Embodied emotion recognition is formulated as a classification task. Given an FPV video, the model is required to predict the overall emotional tendency of the video. Emotions are categorized into positive and negative groups, with intensities ranging from strongly negative (-3) to strongly positive (+3). The dataset consists of 8912 samples for training, 874 for validation, and 891 for testing. The effectiveness of the model is evaluated using average accuracy (Acc) and F1 score.

**Experimental Results.** As shown in Table 2, the first part presents the results of training task-specific models: MMIM [27], CENET [91], and ALMT [103] on $E^3$. The second part presents the zero-shot and fine-tuning (highlighted in grey) results of MLLMs. The experimental results indicate that task-specific models exhibit lower emotion recognition capabilities compared to the zero-shot recognition capabilities of MLLMs, likely due to their limited parameter amount. However, upon fine-tuning, there is a noticeable improvement in the performance of MLLMs. Specifically, VAST [7] outperforms MINI [1] in both zero-shot and fine-tuning scenarios. Our Emotion-LlaMa model, after fine-tuning, achieves the highest performance, highlighting the advantages of incorporating audio features.

## 5.2 Embodied Emotion Classification

**Motivation.** Unlike emotion recognition, which provides an overall emotional assessment, embodied emotion classification focuses on identifying specific emotions of each individual. This detailed approach is essential for applications requiring precise emotional discernment, such as psychotherapy [23, 26, 95, 88] and interactive chatbots [5, 70, 87]. By being able to classify emotions on an individual level, embodied agents can provide personalized responses that are finely tuned to the specific contextual cues of the interaction.

Table 5: Comparing different approaches on **embodied emotion reasoning** task. Our evaluation employs GPT, assessing responses for information correctness (IC), detailed orientation (DO), contextual understanding (CU), and temporal understanding consistency (TUC), alongside standard text quality metrics.

| Methods | IC | DO | CU | TUC | Bleu-3 | Bleu-4 | Rough-L | Cider |
|---------|------|------|------|------|--------|--------|---------|-------|
| HCRN | 0.77 | 1.24 | 1.29 | 0.56 | 8.64 | 5.10 | 27.86 | 17.99 |
| UMT | 0.85 | 1.35 | 1.36 | 0.66 | 8.70 | 5.11 | 28.79 | 18.20 |
| MGN | 0.88 | 1.41 | 1.55 | 0.67 | 9.01 | 5.43 | 28.91 | 18.27 |
| MALMM | 1.31 | 1.32 | 1.74 | 1.51 | 3.12 | 2.98 | 9.74 | 10.67 |
| MALMM | 1.41 | 1.83 | 1.95 | 1.59 | 18.82 | 13.87 | 32.86 | 59.21 |
| VAST | 0.98 | 1.30 | 1.56 | 1.27 | 8.06 | 5.06 | 22.46 | 15.67 |
| VAST | 2.00 | 2.07 | 2.51 | 1.87 | 19.75 | 15.61 | 36.14 | 75.51 |
| MINI | 2.05 | 2.02 | 2.44 | 2.03 | 4.29 | 3.00 | 19.01 | 19.18 |
| MINI | 2.22 | 2.23 | 2.63 | 2.06 | 18.62 | 14.95 | 34.68 | 81.22 |
| Ours | **2.39** | **2.39** | **2.77** | **2.18** | **20.64** | **15.99** | **37.20** | **84.06** |

**Task Definition.** Embodied emotion classification is defined as a closed-set question-and-answer task. Given an FPV video and a query text specifying a person (including the camera wearer), the model is tasked to classify the person's emotion. The model is expected to assign one of eight common emotion categories to the person, each of which is associated with an intensity score ranging from 1 to 3. The dataset comprises 8912 training samples, 900 for validation, and 1472 for testing. The evaluation metrics include the F1 score for each class and the overall Micro-F1 score.

**Experimental Results.** We first evaluate several task-specific models: DFAN [71], VAANet [107] and CTEN [106]. As shown in Table 3, previous task-specific models outperform zero-shot MLLMs but are inferior to fine-tuned MLLMs on the average F1 score. After fine-tuning, MLLMs have enhanced their ability to perceive the emotion of the person in video. Emotion-LlaMa performs almost on par with task-specific models in terms of F1 score. It is worth noting that the Micro-F1 score of MALMM decreased after fine-tuning, due to its long-term memory bank storing reference histories that contradict the predicted target emotions.

## 5.3 Embodied Emotion Localization

**Motivation.** This benchmark aims to identify the precise moments when specific emotions occur in FPV videos. Grasping the timing of emotional occurrences is essential for crafting systems capable of delivering timely and contextually appropriate interventions or responses. This capability is especially important for applications such as interactive storytelling [2, 66, 81], mental health monitoring [31, 79, 85], and assistive technologies [36, 72, 92], where accurately identifying emotional cues can significantly enhance user engagement and the effectiveness of the system.

**Task Definition.** Embodied emotion localization is defined as a regression task. Given an FPV video and a text query specifying a character and an emotion, the model is asked to predict the start and end times of that emotion in the video. The dataset used for this benchmark is the same as that described in Section 5.2. The evaluation metrics are "mIoU" which indicates the mean intersection over union and "R$n$,U@$m$" which represents the percentage of testing samples that have at least one of the top-$n$ results with IoU larger than $m$. More details can be found in Appendix.

**Experimental Results.** As shown in Table 4, the experimental results indicate that directly adapting current localization models, such as MSAT [104] and r2-tuning [50], does not yield satisfactory results, due to the gap between embodied emotion localization and existing activity localization tasks. The majority of MLLMs process video frames by sampling and encoding them into visual tokens, which often leads to a lack of temporal awareness necessary for localization and hinders their application to this task. Therefore, we fine-tune VTG-LLM [25] and TimeChat [75] which specifically design timestamp knowledge encoders for localization. While these methods may initially perform less effectively in zero-shot scenarios compared to task-specific models, they have shown competitive results after fine-tuning, showcasing their potential for emotion localization. In addition, we find that VTG-LLM and TimeChat tend to output longer intervals during zero-shot inference compared to inference after fine-tuning. This is mainly because they use action localization datasets in the pre-training stage, which have longer action time intervals.

Table 6: Ablation study on emotion reasoning.

| Methods | IC | DO | CU | TUC | B-3 | B-4 | R | C |
|---|---|---|---|---|---|---|---|---|
| Ours | 2.39 | 2.39 | 2.77 | 2.18 | 20.64 | 15.99 | 37.20 | 84.06 |
| $-\Delta_{\text{Subtitle}}$ | 2.25 | 2.31 | 2.71 | 2.00 | 20.09 | 15.52 | 36.30 | 81.53 |
| $-\Delta_{\text{Audio}}$ | 2.22 | 2.23 | 2.63 | 2.06 | 18.62 | 14.95 | 34.68 | 81.22 |
| $-\Delta_{\text{Subtitle\&Audio}}$ | 2.03 | 2.19 | 2.58 | 1.92 | 18.48 | 14.52 | 34.82 | 79.91 |

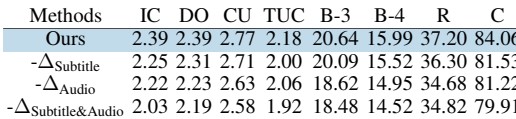 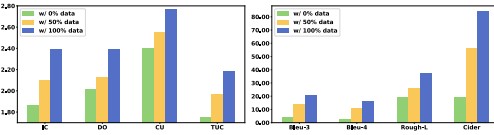

Figure 4: Ablation on training data volume.

## 5.4 Embodied Emotion Reasoning

**Motivation.** This benchmark aims to delve into the root causes of emotions depicted in FPV videos. It surpasses mere detection and classification tasks by striving to infer the underlying reasons behind specific emotions. Exploring the causes of emotions is essential for applications necessitating heightened cognitive empathy, such as counseling bots [34, 59, 82], companion robots [3, 74, 77], and advanced personal assistants [15, 21, 24]. By integrating emotion reasoning capabilities, these systems can provide users with more insightful and contextually relevant services.

**Task Definition.** Embodied emotion reasoning is defined as an open-ended question-and-answer task. The model is presented with an FPV video alongside a textual query, tasked with discerning the underlying reasons behind a specific emotion. For example, a query might prompt, "Why is the girl in black scared?" The answer is required to contain the activity in the video and the cause of the emotion. The dataset utilized for this task is the same as that in Section 5.2. Our evaluation employs GPT, assessing responses for information correctness (IC), detailed orientation (DO), contextual understanding (CU), and temporal understanding consistency (TUC), alongside standard text quality metrics such as Bleu, Rough-L, and Cider. More details can be found in the Appendix.

**Experimental Results.** We first evaluate the performance of prior works: UMT [42], HCRN [40], and MGN [43]. As depicted in the upper block of Table 5, all methods exhibit poor performance, reflecting inherent differences in emotional representation within FPV videos. In contrast, MLLMs exhibit significant performance improvements, particularly in the Cider metric. However, MLLMs still face challenges such as the illusion problem and temporal comprehension issues, which hinder their performance in GPT-related metrics such as IC and TUC.

## 5.5 Ablation Study

In Figure 4 and Table 6, we explore the impact of training data volume and additional modality information on the analysis of embodied emotions. By fine-tuning MLLMs with varying data sizes, our experiments demonstrate that a larger dataset significantly enhances the model's capacity to comprehend embodied emotions. Notably, a marked enhancement is observed when utilizing the complete dataset rather than only 50%, indicating the necessity for further data expansion. Furthermore, we discover that subtitle texts and audio data play pivotal roles, with their combined utilization yielding optimal results due to their complementary nature.

## 6 Conclusion

The $E^3$ dataset represents an advancement in the field of embodied emotion analysis, offering over 70 hours of first-person view (FPV) videos that capture a wide range of human emotions. $E^3$ is featured by its extensive and varied annotations, which not only enrich the dataset but also hold the potential to significantly improve user experiences across a multitude of applications. This robust foundation enables the development of empathetic and context-aware AI agents. Furthermore, our **Emotion-LlaMa** framework, which synergizes visual and acoustic modalities, has proven its efficacy in enhancing the understanding of emotions as conveyed through FPV videos. The comprehensive benchmark tasks established by $E^3$ have set a new benchmark for assessing emotion analysis models. We hope that $E^3$ will catalyze future innovations in multimodal understanding, robotics, and augmented reality, leading to the creation of embodied agents capable of natural human interaction.

## 7 Acknowledgements

This work was supported by the National Natural Science Foundation of China (No.62037001, No.62307032) and the Key Research and Development Program of Zhejiang Province (No. 2023C03192).

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
