# $E^3$: Exploring Embodied Emotion Through A Large-Scale Egocentric Video Dataset

**Wang Lin**[1,*]**, Yueying Feng**[1,*]**, Wenkang Han**[1,*]**, Tao Jin**[1]**, Zhou Zhao**[1]**,**
**Fei Wu**[1]**, Chang Yao**[1]**, Jingyuan Chen**[1,†]**,**
Zhejiang University[1]
{linwanglw, yueyingf, jint_zju, zhaozhou,
wufei, changy, jingyuanchen,}@zju.edu.cn

## Contents

---

[*] Equal Contribution.
[†] Corresponding Author.

38th Conference on Neural Information Processing Systems (NeurIPS 2024) Track on Datasets and Benchmarks.

# A Annotation Details

## A.1 Annotation Process

Prior to conducting the annotation process, human annotators undergo a comprehensive training phase. During this phase, annotators are encouraged to raise any doubts and present corner cases to the first author, who revisits the guidelines with further details and examples. Following the training phase, a meeting is held between the first author, the last author, and the annotation manager to ensure a clear understanding of the task. The manager then evaluates the performance of annotators before collecting the final annotations. Throughout the annotation process, the annotation manager, the first author, and the last author maintain regular communication, reviewing samples and addressing any potential issues such as the exclusion of videos with problematic or offensive content.

## A.2 Distribution of Data Annotators

We employ a total of 26 annotators, consisting of 14 females and 12 males. Among them, 8 hold a bachelor's degree, 12 have attended junior college, and 6 have completed secondary school. The annotators utilize the interface shown in Figure 5 for their work. The pricing for annotations is as follows: emotion recognition is $0.05 per annotation, emotion classification and localization is $0.15 per annotation, and emotion reasoning is $0.20 per annotation. Each annotator can complete 4 tasks for a video within 3 minutes, resulting in an approximate wage of $11 per hour. This rate is about $3 higher than the federal minimum hourly wage in the United States.

# B More Dataset Statictics

## B.1 Data Splits

We randomly split the dataset into training, validation, and test sets in the ratio of $8 : 1 : 1$ as shown in Table 7 and 8. For the embodied emotion recognition task, there are $17,598$ training samples, $2,200$ validation samples, and $2,200$ test samples. The embodied emotion classification, localization, and reasoning tasks use the same dataset, with $15,821$ training samples, $1,977$ validation samples, and $1,977$ test samples.

The discrepancy in the number of samples between the emotion recognition task and the other three tasks is due to some videos having an overall emotional tendency but not being suitable for analyzing individual characters. This can occur in videos with a large number of characters or when annotators are unable to identify the reasons behind a character's emotions.

## B.2 Relationship Between Emotions and Topics

In order to better study the distribution of emotions, we constructed a graph based on the distribution of each emotion across different topics, as shown in Figure 7. *Happy* and *Sad* tend to frequently appear together, while *Scared* tends to appear alone. *Sarcastic* is connected to multiple topics, such as "humor" and "family conflict", indicating that sarcasm is prevalent in these contexts. *Angry* is primarily associated with negative topics, such as "betrayal", "misunderstanding", and "emergencies", demonstrating the prevalence of anger in these situations. Certain topics, such as "family conflict", "friendship", and "fashion", are linked to multiple emotions, suggesting that these topics elicit diverse emotional responses. In contrast, topics like "food" and "shopping" are more often associated with positive emotions (*e.g.*, *Happy* and *Surprised*).

Overall, this relationship graph reveals the complex and diverse patterns of associations between emotions and topics, illustrating how different emotions are represented and significant in each context. This visualization helps us better understand the interplay and interaction of emotions across various topics.

# C Baseline Model Details

In the multimodal domain [47, 61, 62, 63, 93, 94], there are many basic perception and understanding tasks. In this section, we will introduce four related tasks and baseline models in detail.

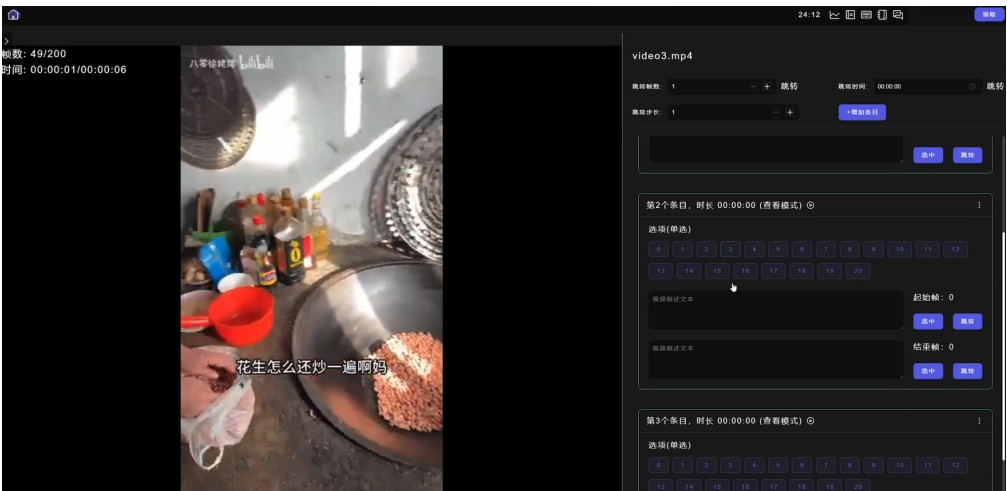

Figure 5: Annotation interface used by the annotator.

Table 7: Data split for Emotion Recognition.

| Split | # of video | # of annotation |
|---|---|---|
| Train | 17598 | 17598 |
| Val | 2200 | 2200 |
| Test | 2200 | 2200 |
| Total | 21998 | 21998 |

Table 8: Data split for Emotion Classification, Localization and Reasoning.

| Split | # of video | # of annotation |
|---|---|---|
| Train | 12269 | 15821 |
| Val | 1913 | 1977 |
| Test | 1906 | 1977 |
| Total | 16088 | 19775 |

## C.1  Baseline for Embodied Emotion Recognition

We compare our method with three task-specific models: MMIM [27], CENET [91], and ALMT [103] and three multimodal large language model: MALMM [28], VAST [7], and MINI [1]. The details of these models are as follows:

- **MMIM [27]:** The MultiModal InfoMax (MMIM) framework enhances multimodal emotion analysis by hierarchically maximizing the Mutual Information (MI) in unimodal input pairs (inter-modality) and between multimodal fusion result and unimodal input in order to maintain task-related information through multimodal fusion. It further formulates a set of computationally simple parametric and non-parametric methods to approximate their truth value.

- **CENET [91]:** The Cross-modal Enhancement Network (CENET) integrates visual and acoustic information into a language model to enhance text representations. It uses a cross-modal enhancement module within a transformer-based pre-trained language model to capture long-range emotional cues from unaligned nonverbal data and introduces a feature transformation strategy to reduce distribution differences between verbal and nonverbal modalities.

- **ALMT [103]:** The Adaptive Language-guided Multimodal Transformer (ALMT) incorporates an adaptive hyper-modality learning module to learn representations that suppress irrelevant or conflicting information from visual and audio features under the guidance of language features at different scales. It achieves this through a multimodal fusion approach that results in a complementary and joint representation for effective emotion analysis.

- **MALMM [28]:** The Memory-Augmented Large Multimodal Model (MALMM) addresses the limitations of processing long videos by existing LLM-based multimodal models. It proposes an online processing approach with a long-term memory bank to store past video information, allowing for efficient and effective long-term video understanding without exceeding context length constraints or GPU memory limits.

- **VAST [7]:** The Vision-Audio-Subtitle-Text omni-modality model (VAST) is designed to understand and process videos by integrating vision, audio, and subtitle modalities with text. It uses a large-scale omni-modality video caption dataset called VAST-27M for training, achieving state-of-the-art results on various cross-modality benchmarks.
- **MINI [1]:** The MiniGPT4-Video extends the capabilities of MiniGPT-v2 for single image understanding to video sequences. It processes both temporal visual data and textual conversations, allowing the model to answer queries involving both components effectively, and outperforming existing methods on multiple benchmarks.

## C.2 Baseline for Embodied Emotion Classification

We compare our method with three task-specific models: DFAN [71], VAANet [107] and CTEN [106] and three multimodal large language models as described in embodied emotion recognition. The details of these models are as follows:

- **DFAN [71]:** The Dual Focus Attention Network (DFAN) mimics the human process of emotion recognition in videos by extracting features such as action, object, and scene. It employs two distinct attention modules, Time Series Focus and Frame Objects Focus, to concentrate on temporal and spatial visual cues respectively, effectively identifying the most emotionally charged frames and visual elements.
- **VAANet [107]:** The Visual-Audio Attention Network (VAANet) is an end-to-end approach for recognizing emotions in user-generated videos. It integrates spatial, channel-wise, and temporal attentions into a visual 3D CNN and temporal attentions into an audio 2D CNN. Additionally, VAANet utilizes a novel polarity-consistent cross-entropy loss to guide attention generation, based on the polarity-emotion hierarchy constraint.
- **CTEN [106]:** The Cross-Modal Temporal Erasing Network (CTEN) addresses the challenge of weakly supervised video emotion detection and prediction. This network selects keyframes by leveraging intra- and inter-modal relationships and iteratively erases these keyframes to encourage the model to focus on contextual information. CTEN enhances the model's robustness by considering a wider range of cues for recognizing emotions in videos.

## C.3 Baseline for Embodied Emotion Localization

We compare our method with two task-specific models: MSAT [104] and r2-tuning [50] and two multimodal large language models VTG-LLM [25] and TimeChat [75] which are designed for temporal localization. The multimodal large language models used in the other tasks as well as our proposed Emotion-LlaMA are not compared due to the lack of timestamp information for the localization task. The details of these models are as follows:

- **MSAT [104]:** The Multi-stage Aggregated Transformer Network (MSAT) addresses temporal language localization in videos by introducing a novel visual-language transformer backbone. This backbone enables fine-grained visual-language interactions and alignments. Additionally, MSAT incorporates a multi-stage aggregation module that computes stage-specific representations for accurate moment localization, capturing stage-specific information to enhance discriminative power.
- **r2-tuning [50]:** The r2-Tuning is presented as an efficient image-to-video transfer learning framework for video temporal grounding. It utilizes a lightweight Reversed Recurrent (r2) Tuning Block that is recurrently attached to a frozen CLIP model, starting from the last layer and progressively refining spatial-temporal features.
- **VTG-LLM [25]:** The VTG-LLM model focuses on integrating timestamp knowledge into video Large Language Models (LLMs) to enhance video temporal grounding. It introduces a high-quality instruction tuning dataset, VTG-IT-120K, and a novel model design that includes sequence-time embedding, absolute-time tokens, and slot-based token compression to efficiently incorporate timestamp information.
- **TimeChat [75]:** TimeChat is a time-sensitive multimodal large language model designed for long video understanding. It features a timestamp-aware frame encoder and a sliding video Q-Former, which together bind visual content with timestamps and produce variable-length

video token sequences. TimeChat is also trained on an instruction-tuning dataset to improve its instruction-following abilities for tasks like dense video captioning, temporal grounding, and highlight detection.

## C.4 Baseline for Embodied Emotion Reasoning

We compare our method with three task-specific models: UMT [42], HCRN [40], and MGN [43] and three multimodal large language models as described in embodied emotion recognition. The details of these models are as follows:

- **UMT [42]:** The Temporal-Sensitive Video Foundation Models (VFMs) is a training-efficient approach. It masks out most of the low-semantics video tokens while selectively aligning the unmasked tokens with an Image Foundation Model (IFM) serving as the UnMasked Teacher (UMT), providing semantic guidance for faster convergence and multimodal friendliness.

- **HCRN [40]:** The model introduces a general-purpose reusable neural unit called Conditional Relation Network (CRN) for constructing sophisticated structures for representation and reasoning over video. CRNs support high-order relational and multi-step reasoning, forming a hierarchical architecture that shares the same question as the contextual condition.

- **MGN [43]:** The Multi-Granularity Relational Attention Network (MGN) is designed for Audio-Visual Question Answering that incorporates raw audio. The model features a pairwise potential attention mechanism for local sequential representation and a novel ternary potential attention mechanism for global multimodal representation.

# D Implementation Details

Our visual backbone is EVA-CLIP [83], and we kept the weights frozen. Importantly, we trained the linear projection layer and efficiently fine-tuned the language model using LoRA [29]. Specifically, we fine-tuned the $W_q$ and $W_v$ components with a rank ($r$) of 64 and a LoRA-alpha value of 16. We trained the entire model with a consistent image resolution of $224 \times 224$ pixels to ensure uniformity throughout all stages.

For the audio encoder, we used a vanilla 12-layer ViT-B as the Transformer encoder. We converted the raw waveform, pre-processed as a mono channel under a sampling rate of $16,000$, into 128 Kaldi [69]-compatible Mel-frequency bands using a 25ms Hanning window that shifts every 10ms. For patch embedding, we employed convolutional kernels with a size of $(16, 16)$ and stride in time and frequency, ensuring non-overlapping patches to avoid shortcuts through overlap in self-supervision.

We set a batch size of 4 and used the AdamW optimizer along with a cosine learning rate scheduler, setting the learning rate to 1e-4. It took 40 hours on two Nvidia A800 GPUs to fine-tune Emotion-LlaMa on the $E^3$ dataset. The implementation details of the other baseline models used in our experiments are consistent with those in the original paper mentioned in Section C.

# E Qualitative Experimental Results

## E.1 Results of Embodied Emotion Recognition

Figure 8 shows the embodied emotion recognition results of different methods in two cases. Each case contains multiple keyframes and compares the recognition results of several methods, including our proposed Emotion-LlaMa. For the emotion recognition task, MALMM performs slightly worse, whereas the rest of the task-specific and large models demonstrate high accuracy.

## E.2 Results of Embodied Emotion Classification

Figure 9 shows the embodied emotion classification results of different methods in two cases. In the first case, the question is about the boy's emotions. While the ground truth (Gth) labels the emotion as "surprised", DFAN identified it as "angry". Most zero-shot methods, including MALMM, VAST, and MINI, classified the emotion as "happy". However, Emotion-LlaMa accurately identified the boy's emotion as "surprised", matching the ground truth. This is because Emotion-LlaMa introduces

an audio branch, where the emotion of surprise has more distinctive features in audio than in visuals such as facial expressions and body movements, preventing the model from incorrectly categorizing it as "angry" or "happy".

### E.3 Results of Embodied Emotion Localization

Figure 10 shows the embodied emotion localization results of different methods in two cases. In both cases, task-specific methods such as MSAT and r2-tuning identify time intervals close to the true value. In contrast, the multimodal large language model predicts a wider range of times at zero-shot, which is refined after fine-tuning. For example, in Figure 10(a), VTG-LLM (Zero Shot) identifies the range as 0.0-5.0 seconds and fine-tunes it to 4.0-6.0 seconds. However, overall, the current methods have low accuracy in embodied emotion localization, especially for multimodal large language models, and improving their time-awareness remains a challenge.

### E.4 Results of Embodied Emotion Reasoning

Figure 11 shows the embodied emotion reasoning results of different methods in two cases. HCRN and UMET treat it as a multi-label classification problem and the. MGN employs a decoder to generate the answer. Both of these mothers do not have a high answer accuracy. Multimodal large language models benefit from large-scale pre-training, enhancing their reasoning capabilities. However, while methods such as "zero-shot" and "fine-tuning" provide a range of explanations, the problem of illusions is significant. For instance, in the first case, the zero-shot method focuses on her crying, and in the second case, MALMM answers seeing a lot of money. The experimental results show that Emotion-LlaMa exhibits superior performance in emotional reasoning. In both cases, Emotion-LlaMa's answers closely matched the truth, demonstrating its ability to accurately understand and reason about the causes of emotions.

## F Prompt for GPT evaluation

To more thoroughly evaluate the model's performance for emotion reasoning, we are inspired by [52] and develop an evaluation pipeline using the GPT-3.5 model. This pipeline assesses various capabilities of the model and assigns a relative score to the generated predictions on a scale of $1 - 5$, in the following four aspects:

- **Information Correctness:** We verify the accuracy of the generated text, ensuring it aligns with the video content and doesn't misinterpret or misinform. The prompt is shown in Table 9.

- **Detail Orientation::** We evaluate the depth of the model's responses, looking for both completeness, meaning the model's response covers all major points from the video, and specificity, denoting the inclusion of specific details rather than just generic points in the model's response. The prompt is shown in Table 10.

- **Contextual Understanding:** We assess the model's understanding of the video's context, checking if its responses aligns with the overall context of the video content. The prompt is shown in Table 11.

- **Temporal Understanding Consistency:** We examine the model's grasp of the temporal sequence of events in the video when answering questions. The prompt is shown in Table 12.

We present the results of the evaluation of embodied emotion reasoning in Table 5 in Section 5.4 of the main paper. The results reveal Emotion-LlaMa's competent performance across all key aspects when compared with the recently introduced video understanding models.

## G Privacy Protection

During the construction of the dataset for conducting emotion analysis, we strictly adhere to the principles of privacy protection to ensure the security and anonymity of the personal information of all participants. Below is a list of the measures we took:

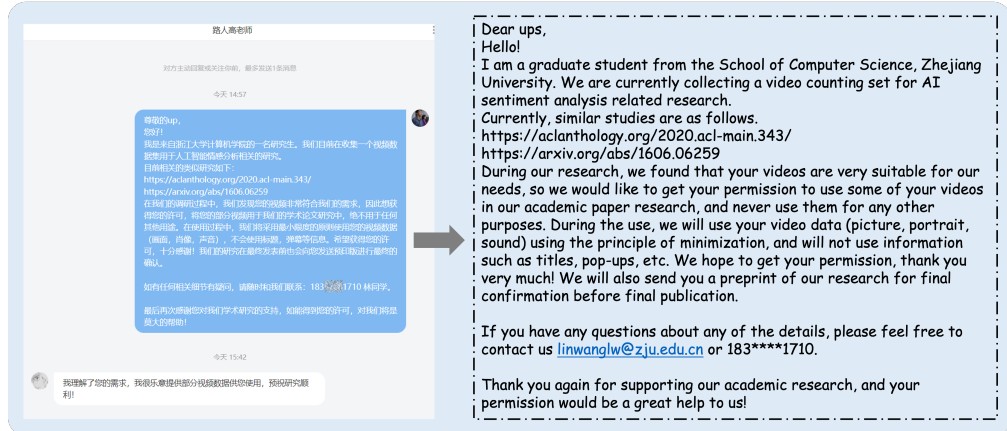

Figure 6: An example of consent for data collection.

- **Consent for Data Collection:** Before collecting any video data, we obtained explicit consent from the video owners. We explained to them the purpose of the study, how the data would be used, and how their personal information would be protected, as shown in Figure 6.
- **Legal Compliance:** We ensure that all data collection and processing activities comply with relevant data protection laws and regulations, including but not limited to YouTube, Bilibili, Douyin's platform terms and conditions, and more.
- **Transparency:** We provide participants with transparent information about how their data is collected, used, stored, and protected, as well as their right to withdraw consent at any time.
- **Data Minimization:** We only collect data that is absolutely necessary for the purpose of the study and promptly delete or destroy it when it is no longer needed.
- **Participants' Rights:** We respect the rights of participants, including the right to access, the right to rectification, and the right to be forgotten. Participants can request to see, correct, or delete their data at any time.
- **Access Controls:** We restrict access to and use of datasets, and researchers are required to adhere to strict data use protocols.

Through these measures, we strive to build a dataset that contributes to emotion analysis research while fully protecting individual privacy.

## H  Limitations

We acknowledge three limitations in our study: bias, subjectivity, and modality.

- **Dataset Bias:** The $E^3$ dataset, although diverse, is collected from specific sources and may not fully represent the global spectrum of emotional expressions and cultural nuances. This potential bias could affect the generalizability of models trained on this dataset.
- **Annotation Subjectivity:** The annotations in $E^3$ rely on human judgment, which is inherently subjective. Different annotators might have varying interpretations of the same emotional expression, introducing inconsistencies in the dataset.
- **Modality Limitations:** Although $E^3$ incorporates visual, acoustic, and textual modalities, it may not capture all aspects of emotional expression, such as physiological signals (*e.g.*, heart rate or body temperature), which could provide additional insights into emotion analysis.

Addressing these limitations will be crucial for future work, ensuring that the $E^3$ dataset and models like Emotion-LlaMa can continue to evolve and meet the demands of advancing emotion analysis and real-world applications.

# I    Societal Impacts

The development and application of the $E^3$ dataset and the Emotion-LlaMa framework carry several societal implications that need to be considered.

The enhanced understanding of human emotions through technology can lead to significant improvements in human-computer interaction, making digital interfaces more empathetic and responsive to user emotions. This advancement can be particularly beneficial in applications related to mental health, where early detection of emotional distress can be facilitated through emotion recognition systems, enabling timely interventions. In the field of education, emotion-aware systems can adapt to the emotional states of students, offering personalized learning experiences that are more engaging and conducive to learning. Moreover, for individuals with disabilities or communication barriers, emotion recognition technologies can provide alternative means of interaction, enhancing their accessibility to digital services and improving their quality of life.

However, the development and application of these technologies also come with challenges. One of the primary concerns is the potential for these systems to perpetuate or exacerbate existing biases. It is essential to ensure that these technologies are designed and continuously refined with fairness in mind, to prevent any unfair treatment of individuals based on ethnicity, gender, age, or other demographic factors. Misinterpretation of emotions is another significant challenge. Automated systems, if not sophisticated enough, may misinterpret emotional cues, leading to incorrect assumptions about an individual's feelings or intentions. The repercussions of such misinterpretations could be profound, resulting in inappropriate responses from machines and

# J    Responsibility & Dataset Liscence

We bear all responsibilities for the content, licensing, distribution, and maintenance of our datasets in $E^3$. Our datasets are released under a CC-BY-4.0 license. Data, code, and annotation guidelines are hosted on GitHub at the following URL: https://exploring-embodied-emotion-official.github.io.

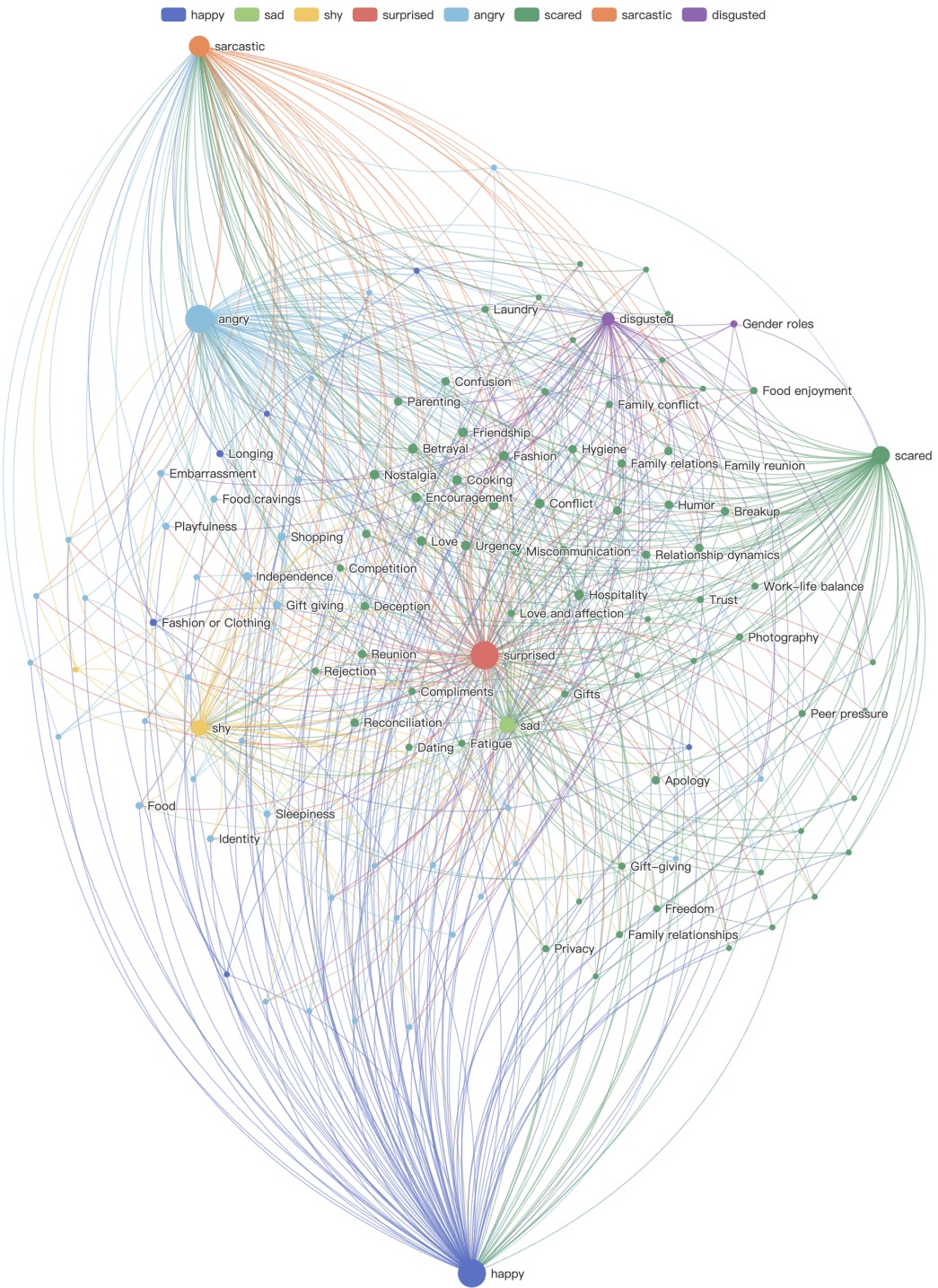

Figure 7: Relationship graph between emotions and topics.

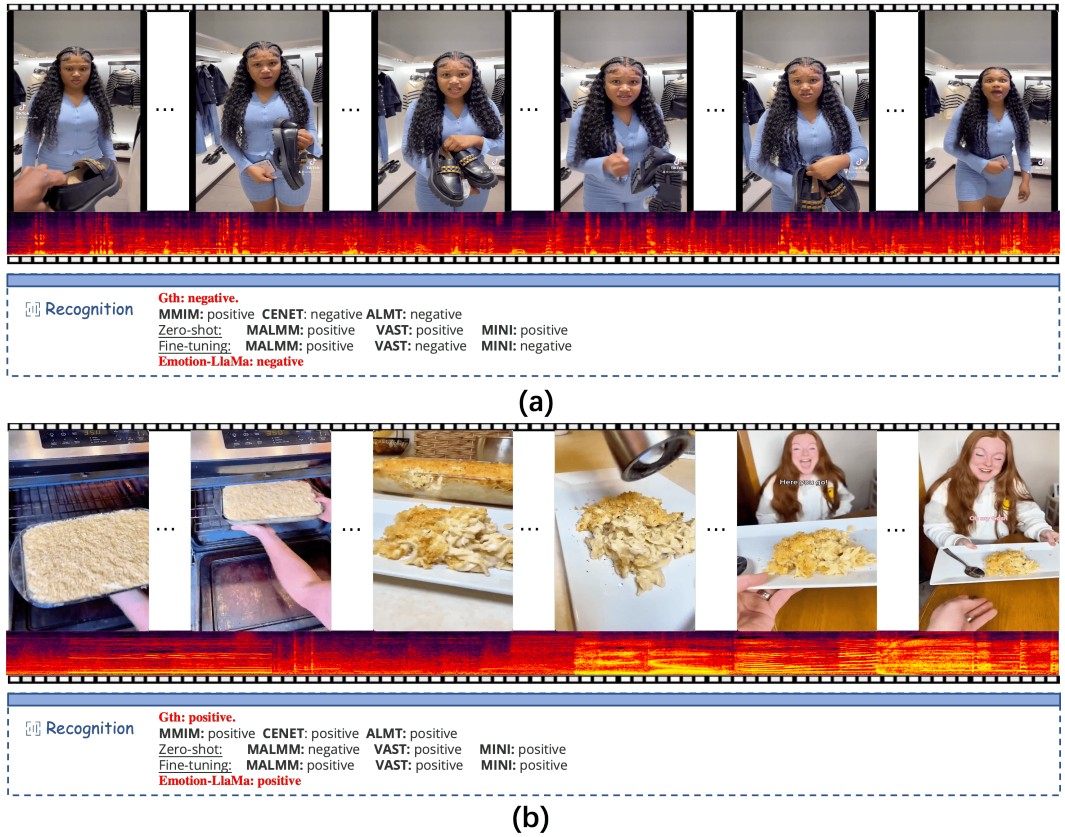

Figure 8: Results of embodied emotion recognition.

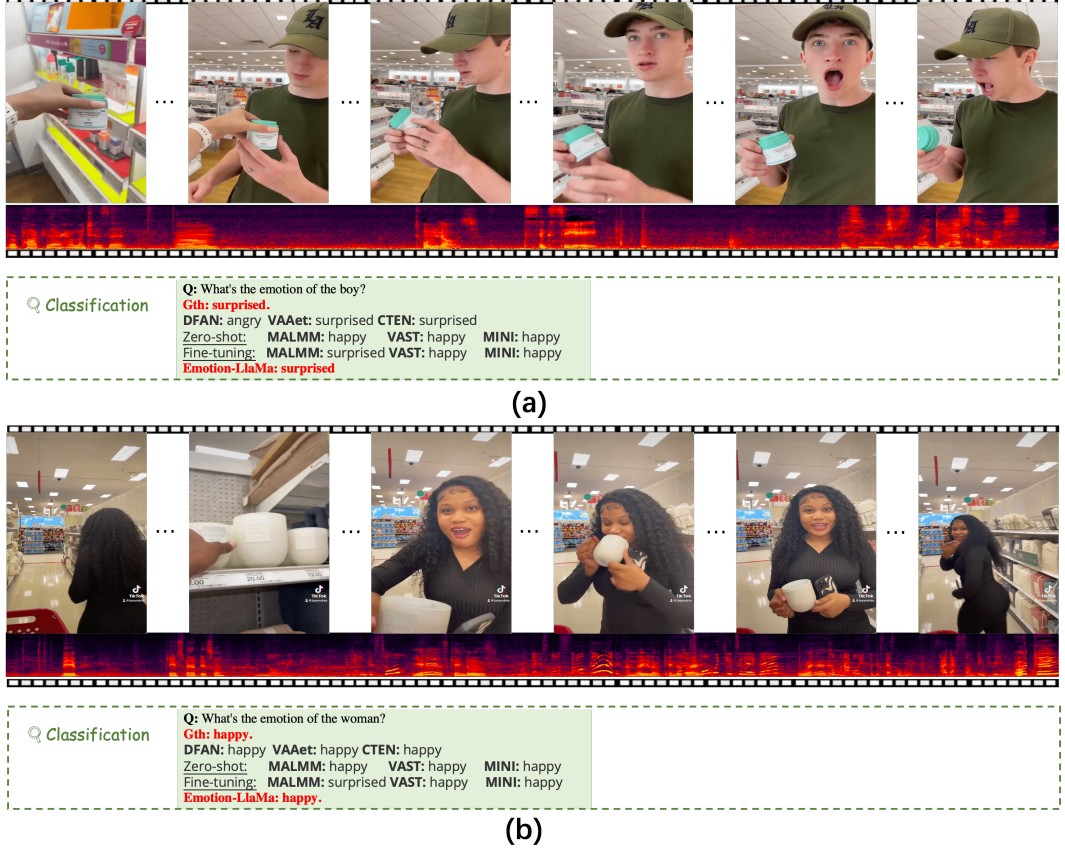

Figure 9: Results of embodied emotion classification.

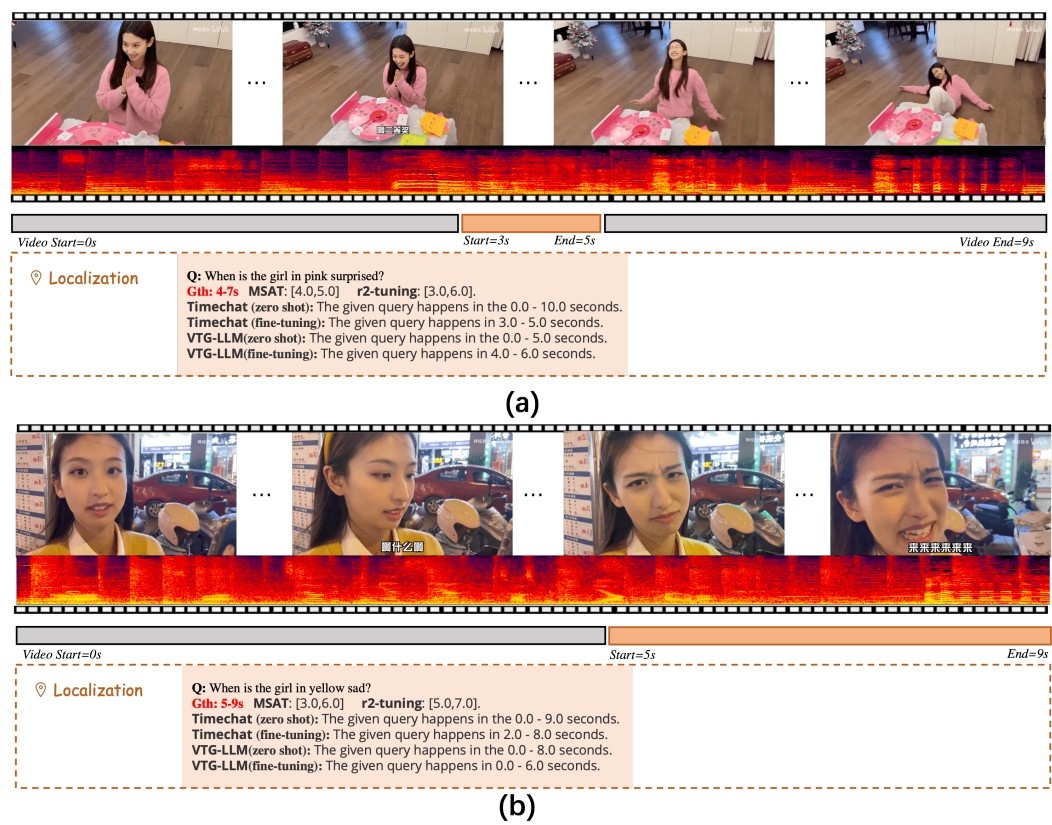

Figure 10: Results of embodied emotion localization.

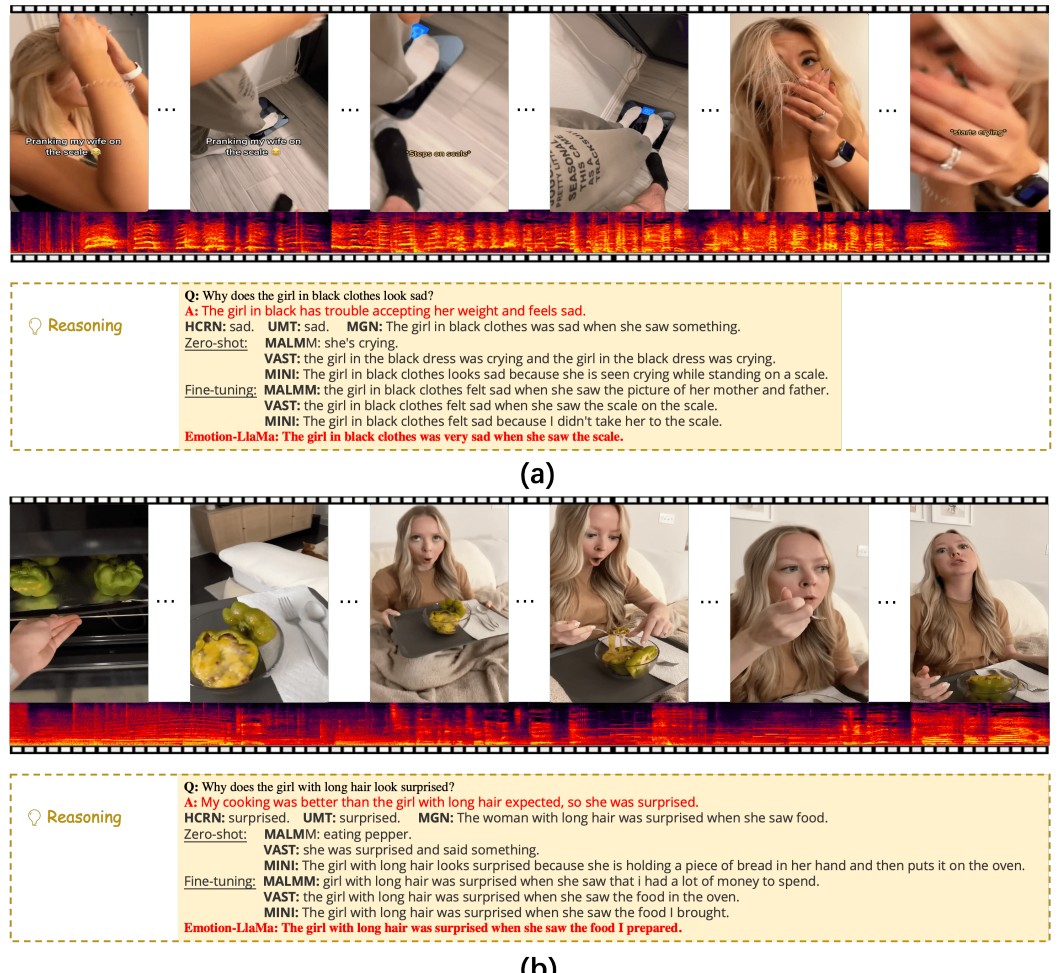

Figure 11: Results of embodied emotion reasoning.

Table 9: Prompt for Information Correctness (IC).

```
{ "role":  "system",
"content":
"You are an intelligent chatbot designed for evaluating the factual
accuracy of generative outputs for video-based question-answer pairs.  "
"Your task is to compare the predicted answer with the correct answer and
determine if they are factually consistent.  Here's how you can accomplish
the task:"
"---"
"##INSTRUCTIONS: "
"- Focus on the factual consistency between the predicted answer and the
correct answer.  The predicted answer should not generalize or contain any
misinterpretations or irrelevant information."
"- The predicted answer must be factually accurate and align with the video
content."
"- Consider synonyms or paraphrases as valid matches."
"- Evaluate the factual accuracy of the prediction compared to the
answer."},
{ "role":  "user",
"content":
"Please evaluate the following video-based question-answer pair:"
f"Question:  question"
f"Correct Answer:  {answer}"
f"Predicted Answer:  {pred}"
"Provide your evaluation only as a factual accuracy score where the factual
accuracy score is an integer value between 0 and 5, with 5 indicating the
highest level of factual consistency.  "
"Please generate the response in the form of a Python dictionary string
with keys 'score', where its value is the factual accuracy score in INTEGER,
not STRING."
"DO NOT PROVIDE ANY OTHER OUTPUT TEXT OR EXPLANATION. Only provide the
Python dictionary string.  "
"For example, your response should look like this:  {'score':  4.8}." }
```

Table 10: Prompt for Detailed Orientation (DO).

```
{ "role":  "system",
"content":
"You are an intelligent chatbot designed for evaluating the detail
orientation of generative outputs for video-based question-answer pairs.
"
"Your task is to compare the predicted answer with the correct answer
and determine its level of detail, considering both completeness and
specificity.  Here's how you can accomplish the task:"
"---"
"##INSTRUCTIONS: "
"- Check if the predicted answer covers all major points from the video.
The response should not leave out any key aspects."
"- Evaluate whether the predicted answer includes specific details rather
than just generic points.  It should provide comprehensive information that
is tied to specific elements of the video."
"- Consider synonyms or paraphrases as valid matches."
"- Provide a single evaluation score that reflects the level of detail
orientation of the prediction, considering both completeness and
specificity."},
{ "role":  "user",
"content":
"Please evaluate the following video-based question-answer pair:"
f"Question:  question"
f"Correct Answer:  {answer}"
f"Predicted Answer:  {pred}"
"Provide your evaluation only as a detail orientation score where the
detail orientation score is an integer value between 0 and 5, with 5
indicating the highest level of detail orientation.  "
"Please generate the response in the form of a Python dictionary string
with keys 'score', where its value is the factual accuracy score in INTEGER,
not STRING."
"DO NOT PROVIDE ANY OTHER OUTPUT TEXT OR EXPLANATION. Only provide the
Python dictionary string.  "
"For example, your response should look like this:  {'score':  4.8}." }
```

Table 11: Prompt for Contextual Understanding (CU).

```
{ "role":  "system",
"content":
"You are an intelligent chatbot designed for evaluating the contextual
understanding of generative outputs for video-based question-answer pairs.
"
"Your task is to compare the predicted answer with the correct answer and
determine if the generated response aligns with the overall context of the
video content.  Here's how you can accomplish the task:"
"---"
"##INSTRUCTIONS: "
"- Evaluate whether the predicted answer aligns with the overall context
of the video content.  It should not provide information that is out of
context or misaligned, nor should it not generalize"
"- The predicted answer must capture the main themes and sentiments of the
video."
"- Consider synonyms or paraphrases as valid matches."
"- Provide your evaluation of the contextual understanding of the
prediction compared to the answer."},
{ "role":  "user",
"content":
"Please evaluate the following video-based question-answer pair:"
f"Question:  question"
f"Correct Answer:  {answer}"
f"Predicted Answer:  {pred}"
"Provide your evaluation only as a contextual understanding score where the
contextual understanding score is an integer value between 0 and 5, with 5
indicating the highest level of contextual understanding.  "
"Please generate the response in the form of a Python dictionary string
with keys 'score', where its value is contextual understanding score in
INTEGER, not STRING."
"DO NOT PROVIDE ANY OTHER OUTPUT TEXT OR EXPLANATION. Only provide the
Python dictionary string.  "
"For example, your response should look like this:  {'score':  4.8}." }
```

Table 12: Prompt for Temporal Understanding Consistency (TUC).

```
{ "role":  "system",
"content":
"YYou are an intelligent chatbot designed for evaluating the temporal
understanding of generative outputs for video-based question-answer pairs.
"
"Your task is to compare the predicted answer with the correct answer and
determine if they correctly reflect the temporal sequence of events in the
video content.  Here's how you can accomplish the task:"
"---"
"##INSTRUCTIONS: "
"- Focus on the temporal consistency between the predicted answer and the
correct answer.  The predicted answer should correctly reflect the sequence
of events or details as they are presented in the video content and should
not generalize."
"- Consider synonyms or paraphrases as valid matches, but only if the
temporal order is maintained."
"- Evaluate the temporal accuracy of the prediction compared to the
answer."},
{ "role":  "user",
"content":
"Please evaluate the following video-based question-answer pair:"
f"Question:  question"
f"Correct Answer:  {answer}"
f"Predicted Answer:  {pred}"
"Provide your evaluation only as a temporal accuracy score where the
temporal accuracy score is an integer value between 0 and 5, with 5
indicating the highest level of temporal consistency.   "
"Please generate the response in the form of a Python dictionary string
with keys 'score', where its value is the temporal accuracy score in
INTEGER, not STRING."
"DO NOT PROVIDE ANY OTHER OUTPUT TEXT OR EXPLANATION. Only provide the
Python dictionary string.   "
"For example, your response should look like this:  {'score':  4.8}." }
```

# K  Datasheet for $E^3$

## K.1  Motivation

- **For what purpose was the dataset created? Was there a specific task in mind?** Was there a specific gap that needed to be filled? Please provide a description.
  Existing emotion datasets still face the following challenges: (1) emotion analysis datasets are mainly third-person view videos, lacking first-person view emotion analysis videos; (2) first-person view videos are mainly activity recognition, lacking emotion; and (3) emotion analysis annotations are only focused on categorization, lacking fine-grained annotations. $E^3$ provides large-scale, diverse, and fine-grained first-person sentiment analysis video data, which fills the gap between emotion analysis and ego-centric video data. It can support emotion recognition, classification, localization, reasoning, and other sentiment analysis tasks. For a more detailed analysis, see the Introduction section.

- **Who created the dataset (e.g., which team, research group) and on behalf of which entity (e.g., company, institution, organization)?**
  AI+X Lab created the dataset on behalf of Zhejiang University.

- **Who funded the creation of the dataset?** If there is an associated grant, please provide the name of the grantor and the grant name and number.
  This work was supported by the National Natural Science Foundation of China under Grants 62037001.

- **Any other comments?**
  No.

## K.2  Composition

- **What do the instances that comprise the dataset represent (e.g., documents, photos, people, countries)?** Are there multiple types of instances (e.g., movies, users, and ratings; people and interactions between them; nodes and edges)? Please provide a description.
  Our dataset consists of 2 key entities (instances). The entities include videos and annotations. For a more in-depth understanding of these components, we recommend referring to Section 3.2 and 3.3.

- **How many instances are there in total (of each type, if appropriate)?**
  $E^3$ includes $21,998$ videos, $21998$ annotations for embodied emotion recognition, $19,750$ for embodied emotion classification, localization and reasoning.

- **Does the dataset contain all possible instances or is it a sample (not necessarily random) of instances from a larger set?** If the dataset is a sample, then what is the larger set? Is the sample representative of the larger set (e.g., geographic coverage)? If so, please describe how this representativeness was validated/verified. If it is not representative of the larger set, please describe why not (e.g., to cover a more diverse range of instances, because instances were withheld or unavailable).
  To improve the diversity of the dataset, we collected videos involving multiple languages such as Chinese, English, and Korean. To improve the quality of the dataset, we filter out videos without emotion, too short in duration, and videos in which the camera wearer is not involved in the video activity while constructing the dataset.

- **What data does each instance consist of?** "Raw" data (e.g., unprocessed text or images) or features? In either case, please provide a description.
  Each video has a video ID. each video has corresponding 4 types of annotations including recognition annotation, classification annotation, localization annotation, and reasoning annotation. Each video has one recognition annotation, but may have multiple classification annotations, localization annotations, and reasoning annotations.

- **Is there a label or target associated with each instance?** If so, please provide a description.
  No.

- **Is any information missing from individual instances?** If so, please provide a description, explaining why this information is missing (e.g., because it was unavailable). This does not include intentionally removed information, but might include, e.g., redacted text.
  No

- **Are relationships between individual instances made explicit (e.g., users' movie ratings, social network links)?** If so, please describe how these relationships are made explicit.
  N/A.

- **Are there recommended data splits (e.g., training, development/validation, testing)?** If so, please provide a description of these splits, explaining the rationale behind them.
  Yes, we recommend referring to Section 5.1, 5.2, 5.3, and 5.4.

- **Are there any errors, sources of noise, or redundancies in the dataset?** If so, please provide a description.
  No.

- **Is the dataset self-contained, or does it link to or otherwise rely on external resources (e.g., websites, tweets, other datasets)?** If it links to or relies on external resources, a) are there guarantees that they will exist, and remain constant, over time; b) are there official archival versions of the complete dataset (i.e., including the external resources as they existed at the time the dataset was created); c) are there any restrictions (e.g., licenses, fees) associated with any of the external resources that might apply to a dataset consumer? Please provide descriptions of all external resources and any restrictions associated with them, as well as links or other access points, as appropriate.
  $E^3$ is self-contained.

- **Does the dataset contain data that might be considered confidential (e.g., data that is protected by legal privilege or by doctor–patient confidentiality, data that includes the content of individuals' non-public communications)?** If so, please provide a description.
  No. We confirmed that the user-generated data was strictly authorized during the registration process, as specified in Section G.

- **Does the dataset contain data that, if viewed directly, might be offensive, insulting, threatening, or might otherwise cause anxiety?** If so, please describe why.
  No.

If the dataset does not relate to people, you may skip the remaining questions in this section.

- **Does the dataset identify any subpopulations (e.g., by age, gender)?** If so, please describe how these subpopulations are identified and provide a description of their respective distributions within the dataset.
  Yes. Identification can be performed by video visual frames and annotation text, *e.g.*, "a girl wearing glasses".

- **Is it possible to identify individuals (i.e., one or more natural persons), either directly or indirectly (i.e., in combination with other data) from the dataset?** If so, please describe how.
  Yes. Due to the need for emotion analysis (*e.g.*, expression and speech), the dataset contains information about faces and voices. So it may be possible to recognize individuals by face or voice. We follow the principle of minimum data usage in the creation of the dataset and do not include other personal information such as ID numbers.

- **Does the dataset contain data that might be considered sensitive in any way (e.g., data that reveals race or ethnic origins, sexual orientations, religious beliefs, political opinions or union memberships, or locations; financial or health data; biometric or genetic data; forms of government identification, such as social security numbers; criminal history)?** If so, please provide a description.
  No.

### K.3 Collection Process

- **How was the data associated with each instance acquired?** Was the data directly observable (e.g., raw text, movie ratings), reported by subjects (e.g., survey responses), or indirectly inferred/derived from other data (e.g., part-of-speech tags, model-based guesses for age or language)? If the data was reported by subjects or indirectly inferred/derived from other data, was the data validated/verified? If so, please describe how.
  The data are manually annotated by the annotators and reviewed by the annotation manager and the authors.

- **What mechanisms or procedures were used to collect the data (e.g., hardware appara-tuses or sensors, manual human curation, software programs, software APIs)?** How were these mechanisms or procedures validated?
  Manual human curation. we recommend referring to Sections A.

- **If the dataset is a sample from a larger set, what was the sampling strategy (e.g., deterministic, probabilistic with specific sampling probabilities)?**
  $E^3$ isn't sampled from a larger set. But we did create an FPV video subdataset with emotion, please refer to Section 3.2.

- **Who was involved in the data collection process (e.g., students, crowdworkers, contrac-tors) and how were they compensated (e.g., how much were crowdworkers paid)?**
  We employ a total of 26 annotators. All annotators can complete 4 tasks for a video within 3 minutes, leading to a wage pay of around 11 dollars per hour. please refer to Section A.2.

- **Over what timeframe was the data collected?** Does this timeframe match the creation timeframe of the data associated with the instances (e.g., recent crawl of old news articles)? If not, please describe the timeframe in which the data associated with the instances was created.
  N/A.

- **Were any ethical review processes conducted (e.g., by an institutional review board)?** If so, please provide a description of these review processes, including the outcomes, as well as a link or other access point to any supporting documentation.
  N/A.

If the dataset does not relate to people, you may skip the remaining questions in this section.

- **Did you collect the data from the individuals in question directly, or obtain it via third parties or other sources (e.g., websites)?**
  The videos are collected from 3 popular websites: `YouTube`, `BiliBili` and `Douyin`.

- **Were the individuals in question notified about the data collection?** If so, please describe (or show with screenshots or other information) how notice was provided, and provide a link or other access point to, or otherwise reproduce, the exact language of the notification itself.
  We confirmed that the user-generated data was strictly authorized during the registration process, as specified in Section G.

- **Did the individuals in question consent to the collection and use of their data?** If so, please describe (or show with screenshots or other information) how consent was requested and provided, and provide a link or other access point to, or otherwise reproduce, the exact language to which the individuals consented.
  N/A.

- **If consent was obtained, were the consenting individuals provided with a mechanism to revoke their consent in the future or for certain uses?** If so, please provide a description, as well as a link or other access point to the mechanism (if appropriate).
  Yes. Please refer to Section G.

- **Has an analysis of the potential impact of the dataset and its use on data subjects (e.g., a data protection impact analysis) been conducted?** If so, please provide a description of this analysis, including the outcomes, as well as a link or other access point to any supporting documentation.
  Yes. Please refer to Section I.

- **Any other comments?**
  No.

### K.4 Preprocessing/cleaning/labeling

- **Was any preprocessing/cleaning/labeling of the data done (e.g., discretization or bucket-ing, tokenization, part-of-speech tagging, SIFT feature extraction, removal of instances, processing of missing values)?** If so, please provide a description. If not, you may skip the remaining questions in this section.
  We segmented the collected videos based on shot detection for easy annotation. The seg-mentation is seamless and can be easily restored to the original video. The videos were

screened by an annotator to remove emotionless and non-first-person videos. For more detailed information, please refer to Section 3.1.

- **Was the "raw" data saved in addition to the preprocessed/cleaned/labeled data (e.g., to support unanticipated future uses)?** If so, please provide a link or other access point to the "raw" data.
  Yes, the original video can be easily restored by the index label.

- **Is the software that was used to preprocess/clean/label the data available?** If so, please provide a link or other access point.
  N/A.

- **Any other comments?**
  No.

## K.5 Uses

- **Has the dataset been used for any tasks already?** If so, please provide a description.
  Yes, $E^3$ has been used for Embodied Emotion Recognition, Classification, Localization and Reasoning in this current paper.

- **Is there a repository that links to any or all papers or systems that use the dataset?** If so, please provide a link or other access point.
  The current paper and the code used for experiments are available at `https://exploring-embodied-emotion-official.github.io/`.

- **What (other) tasks could the dataset be used for?**
  It can be used for video captioning, action detection, and other video understanding tasks.

- **Is there anything about the composition of the dataset or the way it was collected and preprocessed/cleaned/labeled that might impact future uses?** For example, is there anything that a dataset consumer might need to know to avoid uses that could result in unfair treatment of individuals or groups (e.g., stereotyping, quality of service issues) or other risks or harms (e.g., legal risks, financial harms)? If so, please provide a description. Is there anything a dataset consumer could do to mitigate these risks or harms?
  No.

- **Are there tasks for which the dataset should not be used?** If so, please provide a description.
  N/A.

- **Any other comments?**
  No.

## K.6 Distribution

- **Will the dataset be distributed to third parties outside of the entity (e.g., company, institution, organization) on behalf of which the dataset was created?** If so, please provide a description.
  Yes. The dataset will be publicly available under CC-BY-SA 4.0 license.

- **How will the dataset will be distributed (e.g., tarball on website, API, GitHub)?** Does the dataset have a digital object identifier (DOI)?
  It will be distributed through GitHub.

- **When will the dataset be distributed?**
  Starting from June 2024, the dataset will be distributed to interested parties.

- **Will the dataset be distributed under a copyright or other intellectual property (IP) license, and/or under applicable terms of use (ToU)?** If so, please describe this license and/or ToU, and provide a link or other access point to, or otherwise reproduce, any relevant licensing terms or ToU, as well as any fees associated with these restrictions.
  $E^3$ is licensed under CC-BY-SA 4.0. See `https://github.com/Exploring-Embodied-Emotion-official/E3/blob/main/LICENSE`

- **Have any third parties imposed IP-based or other restrictions on the data associated with the instances?** If so, please describe these restrictions, and provide a link or other

access point to, or otherwise reproduce, any relevant licensing terms, as well as any fees associated with these restrictions.
No.

- **Do any export controls or other regulatory restrictions apply to the dataset or to individual instances?** If so, please describe these restrictions, and provide a link or other access point to, or otherwise reproduce, any supporting documentation.
No.

- **Any other comments?**
No.

### K.7   Maintenance

- **Who will be supporting/hosting/maintaining the dataset?**
Wang Lin and Yueying Feng are supporting/maintaining the dataset.

- **How can the owner/curator/manager of the dataset be contacted (e.g., email address)?**
To contact the authors, kindly send an email to `linwanglw@zju.edu.cn` and `jingyuanchen@zju.edu.cn`.

- **Is there an erratum?** If so, please provide a link or other access point.
No.

- **Will the dataset be updated (e.g., to correct labeling errors, add new instances, delete instances)?** If so, please describe how often, by whom, and how updates will be communicated to dataset consumers (e.g., mailing list, GitHub)?
N/A.

- **If the dataset relates to people, are there applicable limits on the retention of the data associated with the instances (e.g., were the individuals in question told that their data would be retained for a fixed period of time and then deleted)?** If so, please describe these limits and explain how they will be enforced.
No.

- **Will older versions of the dataset continue to be supported/hosted/maintained?** If so, please describe how. If not, please describe how its obsolescence will be communicated to dataset consumers.
N/A.

- **If others want to extend/augment/build on/contribute to the dataset, is there a mechanism for them to do so?** If so, please provide a description. Will these contributions be validated/verified? If so, please describe how. If not, why not? Is there a process for communicating/distributing these contributions to dataset consumers? If so, please provide a description.
Any modification and extension of the dataset under CC-BY-SA 4.0 license is permitted.

- **Any other comments?**
No.