# OpenReview forum: "$E^3$: Exploring Embodied Emotion Through A Large-Scale Egocentric Video Dataset"
_NeurIPS.cc/2024/Datasets_and_Benchmarks_Track — NeurIPS 2024 Track Datasets and Benchmarks Poster_

### Official Review · Reviewer_gGZu · 2024-07-23
**Embodied Emotion from Egocentric Video Dataset**

**Rating:** 7
**Confidence:** 4
**Correctness:** Correct.
**Clarity:** The article is clearly described.

**Review:**

This work creates a large-scale egocentric video emotion dataset that aids in the further emotional understanding of interactive scenarios.

Pros:
a. The first large-scale egocentric visual emotion dataset.
b. Multiple emotion benchmark labels: emotion recognition, emotion classification, emotion localization, and emotion reasoning.
Cons:
a. The dataset uses egocentric video data as emotion data, which can enhance the emotional understanding of embodied videos. However, the emotion labels are annotated by third-person crowdsourced annotators, which introduces potential emotion uncertainty. Of course, obtaining accurate first-person perspective labels is challenging, and the crowdsourcing annotation method employed in the paper can reduce this potential uncertainty.
b. In the emotion recognition category, the dataset classifies emotions into two dimensions: negative and positive. This granularity is coarse and cannot express most emotional categories in real scenarios. A more precise emotion category could be the Valence-Arousal two dimensions or the Valence-Arousal-Dominance three dimensions, but this would require higher standards from annotators. This is just an area for improvement for this dataset and does not affect its contribution.

**Strengths:**

a. The first large-scale egocentric visual emotion dataset.
b. Multiple emotion benchmark labels: emotion recognition, emotion classification, emotion localization, and emotion reasoning.

**Additional Feedback:**

I have no further feedback.

**Documentation:**

The paper provides sufficient detail on the data.

**Ethics:**

I have no apparent ethical concerns regarding this paper.

**Limitations:**

See review.

**Opportunities For Improvement:**

Emotion categories can be further refined.

**Relation To Prior Work:**

Adequate.

**Summary And Contributions:**

This work creates the E3 dataset, a large-scale egocentric video dataset with various emotion benchmark tasks, including emotion recognition, emotion classification, emotion localization, and emotion reasoning. It is highly beneficial for understanding emotions in interactions and makes a clear contribution to the field of affective computing. Additionally, this work proposes the Emotion-LlaMa model based on a large language model, which demonstrates excellent emotion analysis performance. Comparatively, the contribution of the dataset outweighs that of the proposed model. There are some areas for improvement in this work, but they do not detract from the dataset's contribution. In summary, the release of this dataset makes a clear contribution to the understanding of emotions in interactive scenarios.

---

> ### Author Rebuttal · Authors · 2024-08-17
>
> ## **Reviewer gGZu**
>
> We are sincerely grateful to the reviewers for dedicating their time and effort to review our work, and we appreciate the recognition of the novelty of our dataset.  We will try to address  reviewer's comments in detail below.
>
> ### **Q1 Annotated by third-person crowdsourced annotators**
>
> For data annotation accuracy, we take the following steps to reduce potential uncertainty in the annotation process
>
> 1.  We first **control the ratio of male to female annotators** to fully reflect the different genders' understanding of emotions.  (see Appendix A.2)
> 2.  Before annotation: Before annotation work starts, annotators are **required to watch complete video instances** with different emotions and different emotion levels to ensure annotation consistency as much as possible.
> 3.  Annotation process: Considering the inevitable subjective factors, we use **three annotators to annotate the same video**.  For emotion categories, if two or more of the three annotators agree on the annotation of an emotion category, we take that emotion category as the final annotation result.  For the emotion level, their average score is taken as the final annotation.
> 4.  Sampling inspection: To further ensure the quality of the video and the accuracy of the annotation, we also perform sampling inspection on the annotated data, and **the sampling acceptance rate of emotion recognition and emotion classification is 98%**.
>
> ### **Q2 Emotion category**
>
> Thanks for your constructive comments.  For the granularity of emotion categorization, there are two main ways to describe emotion space in the field of emotion analysis: discrete emotion space (happy, sad, etc.) and dimensional emotion space (valence-arousal or valence-arousal-dominance three-dimensional categorization).
>
> However, we currently choose to use discrete emotion categorization for the following reasons:
>
> ​            1.     Continuous dimensional classification is more appropriate when **dealing with data sets containing physiological signals (e.g., EEG, ECG, GSR)**.  These physiological signals typically reflect continuous changes in affective states and are able to capture subtle changes in affective dimensions such as valence, arousal, and dominance, see [1] section 12.
>
> ​            2.    Our dataset contains mainly visual, audio, and textual signals.  These signals are more suitable for discrete emotion classification because they are able to capture salient features of specific emotions such as happiness, anger, sadness, etc.  Therefore, the **discrete classification method is more applicable in our study** and can effectively reflect the emotional states conveyed by these signals, see [1] section 12.
>
> ​            3.     At the same time, the discrete emotion space is **more feasible in practice**, considering the ability of the annotators and our requirement of consistency in labeling the datasets.
>
> Of course, we also recognize that the dimensional emotion space can provide a more fine-grained expression of emotion, and in our future work, we will further explore and experiment with higher dimensional emotion annotation methods to continuously improve and expand the performance and application scope of the dataset.
>
>
>
> [1] Emotion recognition and artificial intelligence: A systematic review (2014–2023) and research recommendations

---

> > ### Author Response · Authors · 2024-08-30
> >
> > Dear Reviewer gGZu:
> >
> > We are grateful that this paper has overall been well-received by the reviewers. As the end of the discussion period is approaching, we kindly ask you to acknowledge if our rebuttal has addressed your concerns and give us an opportunity to address any further follow-up. Thank you again for your participation.
> >
> > Best
> > Authors

---

### Official Review · Reviewer_12eZ · 2024-07-25
**This is a dataset paper aimed at addressing issues related to embodied agent emotion.**

**Rating:** 6
**Confidence:** 4
**Clarity:** The paper is well-written.

**Review:**

This paper proposes that the FPV dataset helps embodied agents interact naturally and empathetically. The dataset is notable for the number of modalities it includes—visual, acoustic, and textual—and for its data diversity.  However, there are still some issues with the dataset's annotation and the effectiveness of Emotion-LlaMA:
- Each emotion was assigned an intensity level, but the article does not provide a precise definition or examples of this intensity level, making it difficult to ensure the reasonableness of such an intensity level.
- The paper also does not provide a clear definition of positive and negative groups in Emotion Recognition. What is their relationship with the eight emotion categories? When an individual's specific emotions in a scene reflect an emotional tendency of thought, how is it determined whether the overall emotion is positive or negative?
-  Audio can indeed provide additional emotion information, but converting the entire audio to audio features in the language space of LLMs, does this approach help with the emotion comprehension of a single individual in the video? Additionally, what is the role of audio tokens without temporal information in emotion classification?
- In the first embodied emotion recognition task, did both task-specific models and zero-shot and fine-tuning MLLMs utilize audio features?

**Strengths:**

- It provides a large, FPV-specific dataset for studying embodied emotion.
- It introduces four benchmarking algorithms, three of which outperform previous methods.
- It presents a novel multimodal Large Language Model in the field.

**Additional Feedback:**

There is no additional feedback.

**Correctness:**

Are the claims made in the submission correct? -Yes
If the submission is a dataset, it is constructed in a sound way? -Yes
If it is a benchmark, are the evaluation methods and experiment design appropriate and performed correctly?  -Yes

**Documentation:**

The paper has sufficient detail on data collection and organization,  availability and maintenance, and ethical and responsible use.
The benchmarks are reproducible.

**Ethics:**

No.

**Limitations:**

In the paper, the authors discuss limitations in terms of dataset bias, annotation subjectivity, and modality limitations. This is comprehensive.

**Opportunities For Improvement:**

In the data annotation process, it is not sufficient to simply provide an intensity range and ask annotators to assign scores. Instead, there should be either precise definitions for each intensity level or examples for each intensity, to help annotators learn and understand the value of each intensity level.

**Relation To Prior Work:**

The paper has discussed the difference.

**Summary And Contributions:**

The objective of this work is to further research on emotion analysis from a first-person view (FPV) and to offer robust support for the emotional interaction capabilities of embodied agents. This work makes three key contributions. First, it introduces the E^3 dataset, a collection of FPV videos that capture people's activities and emotions. Second, it establishes four benchmark tasks based on this dataset. Third, it develops a video understanding framework called Emotion-LlaMA.

---

> ### Author Rebuttal · Authors · 2024-08-17
>
> ## **Reviewer 12eZ**
>
> We appreciate your time and effort in reviewing our work, and we have carefully considered your comments. We will be sure to incorporate your suggestions to enhance the overall quality of the paper. We hope the following clarifications can address the reviewer's concerns:
>
> ### **Q1 About the intensity level**
>
> ​            1、     For intensity levels, we follow **previous work **(CMU-MOSEAS[1],CMU-MOSEI[2], MELD[3]) and annotate on **a [0,3] Likert scale[2]** for presence of motion x: [0: no evidenceof x, 1: weakly x, 2: x, 3: highly x].
>
> ​            2、     Before the annotation process began, we do provided annotators with **the same video examples** illustrating specific levels of emotion intensity to **ensure consistent understanding** and application of these levels. We will **include specific examples** of emotion intensity in the Appendix of the final version of the paper to enhance clarity and rigor.
>
> ​            3、     We also try to ensure the accuracy of the intensity score annotation. Each emotion's intensity score is **labeled by 3 annotators**, and their average score is used as the final intensity score. We also control the ratio of male to female annotators to **minimize gender bias** in emotion understanding.
>
> [1] CMU-MOSEAS: A Multimodal Language Dataset for Spanish, Portuguese, German and French
>
> [2] Multimodal Language Analysis in the Wild: CMU-MOSEI Dataset and Interpretable Dynamic Fusion Graph
>
> [3] MELD: A Multimodal Multi-Party Dataset for Emotion Recognition in Conversations
>
> ### **Q2 Definition of positive and negative**
>
> For the overall emotional tendency of the video, we also annotate following the setting of previous work[1]. The purpose of doing this is that we want agents to be able to judge the emotional climate at social events and take the right actions, such as not spoiling a party.
>
> 1）The eight emotion categories are used to determine an individual's emotion category in a clip from an individual's perspective; whereas positive and negative are used to measure the emotional atmosphere of the whole video from the perspective of the whole video.
>
> 2）Specifically, if there is only the camera wearer in the video, the overall emotion of the video corresponds to the individual emotion. The positive videos tend to contain emotions such as happy, surprise, and shyness, and negative videos tend to contain anger, sarcasm, fear, sadness, and disgust.
>
> 3）For videos containing multi-person interaction and similar emotion types, annotators are required to annotate the emotions in accordance with the positive and negative emotions experienced by the camera wearer. In our dataset, there are indeed cases of conflicting emotions of individuals, such as the video of the prank shown in Figure 1 of the paper, where the emotions of the girl in the video are anger and fear, and the boy's emotion is happiness, but the overall emotional atmosphere of the video is lively and positive.
>
> [1] Ch-sims: A Chinese multimodal sentiment analysis dataset with fine-grained annotation of modality.
>
> ### **Q3 About the audio features** **for** **individual**
>
> We complemented the ablation experiments with audio features in embodied emotion classification and reasoning tasks, which aim at understanding individual emotions.
>
> As the results of **Table 2 in rebuttal PDF 3**, **audio features can indeed contribute** to the understanding of individual emotions.
>
>  Audio features can provide quite recognizable features, such as an **angry man's voice, a joyful child's voice, and a scared woman's scream**, and these unique pitch and energy features can help in understanding the individual's emotions.
>
> On the other hand, compared with VAST (which introduces audio description) in Table 5 and Table 3 in the main paper, our model can achieve better results by directly encoding audio features, which is due to the fact that audio description **introduces noise (e.g., hallucinatory text) and loses fine-grained information such as rhythms.**
>
> ### **Q4 Audio features in embodied emotion classification task**
>
> As mentioned in Q3, audio features are mainly used as **a discriminative feature/context to classify** the emotion of a video (such as an angry man's voice, a joyful child's voice, and a scared woman's scream). We acknowledge that temporal information, whether in audio or visual modalities, is crucial for video understanding. However, to the best of our knowledge, current multimodal large models do not have a suitable way to understand and align temporal information in visual or audio modalities. Table 4 in the main paper VTG-LLM and TimeChat for emotion localization are preliminary attempts, but it is still a big challenge and we leave it for future work.
>
>
>
> ### **Q5 Audio features in embodied emotion recognition task**
>
> In the first emotion recognition task, audio information is **used in all models except MA-LLM and MiniGPT4-Video**.
>
> 1) For task-specific models, MMIM uses audio features extracted from COVAREP[1] and P2FA[2]. CENET uses audio features extracted by COVAREP software[3]. ALMT uses audio features extracted by Librosa[4].
>
> 2) For large language models, VAST introduces an audio descriptor to describe audio information as text into the large language model (MiniGPT4-Video uses ASR transcribed text information).
>
> 3) For our Emotion-LlaMa, **one of the first attempts** to introduce audio features into a multimodal large language model, we used an audio encoder, AudioMAE[], but there is still much room for exploration in this area, e.g., subject-separated audio encoding, audio-visual representational learning.
>
> [1] Covarep—a collaborative voice analysis repository for speech technologies
>
> [2] Speaker identification on the scotus corpus.
>
> [3] Covarep: A collaborative voice analysis repository for speech technologies
>
> [4] Libros: Audio and music signal analysis in python
>
> [5] Masked autoencoders that listen

---

> > ### Author Response · Authors · 2024-08-30
> >
> > Dear Reviewer 12eZ:
> >
> > We are grateful that this paper has overall been well-received by the reviewers. As the end of the discussion period is approaching, we kindly ask you to acknowledge if our rebuttal has addressed your concerns and give us an opportunity to address any further follow-up. Thank you again for your participation.
> >
> > Best
> > Authors

---

### Official Review · Reviewer_AVha · 2024-07-25

**Rating:** 8
**Confidence:** 5
**Correctness:** yes
**Clarity:** yes

**Review:**

It introduces E3, the first massive first-person view video dataset for Exploring Embodied Emotion, which contains over 50 hours of video capturing 8 emotion types. The dataset is recorded daily and has various modalities and annotations, defining 4 core benchmark tasks. It also presents Emotion-LlaMa which combines visual and acoustic modalities. The paper is well-written and organized.

+This dataset is the first dataset for embody emotion analysis, which is meaningful for the emotion community.

+This dataset is suitable for four benchmark tasks: Emotion Recognition, Emotion Classification, Emotion Localization, and Emotion Reasoning.

+It presents a baseline framework Emotion-LlaMa.

-The videos are collected from 3 popular websites: YouTube, BiliBili and Douyin. Why not directly use Ego-4D? Meanwhile, how to ensure these videos are eccentric?

-There is a lack of analysis of modality combination.

**Strengths:**

This paper presents the first massive first-person view video dataset for Exploring Embodied Emotion, which is significant for affective computing and embodied AI communities. Meanwhile, it introduces a baseline to the analysis of the dataset.

**Additional Feedback:**

no

**Documentation:**

yes

**Limitations:**

yes

**Opportunities For Improvement:**

-The videos are collected from 3 popular websites: YouTube, BiliBili and Douyin. Why not directly use Ego-4D? Meanwhile, how to ensure these videos are eccentric?

-There is a lack of analysis of modality combination.

**Relation To Prior Work:**

yes

**Summary And Contributions:**

This paper introduces E3, the first massive first-person view video dataset for exploring embodied emotion, which contains over 50 hours of video capturing 8 emotion types. The dataset is recorded in daily life and has various modalities and annotations, defining 4 core benchmark tasks. It also presents Emotion-LlaMa which combines visual and acoustic modalities.

---

> ### Author Rebuttal · Authors · 2024-08-17
>
> ## **Reviewer AVha**
>
> We are sincerely grateful to the reviewers for dedicating their time and effort to review our work, and we appreciate the recognition of the novelty of our dataset. We will try to address  reviewer's comments in detail below.
>
> ### **Q1 Why not directly use Ego-4D?**
>
> We considered using the Ego-4D dataset at the beginning of the dataset construction. However, through detailed investigation and analysis, we found that the Ego-4D dataset was unsuitable in the following aspects, and thus finally chose the self-collected data source.
>
> 1. **Insufficient richness of human interaction scenes**: Although Ego-4D covers a wide range of daily activities and scenes, the scenes of interaction with people are relatively limited, focusing mainly on a few scenes such as playing poker and family gatherings. The content of these scenarios is relatively homogeneous and cannot meet our data set richness requirements.
> 2. **Lack of emotional expression**: We found that in Ego-4D, even in scenes where there are interpersonal interactions, the emotions of the participants are usually neutral and lack obvious emotional fluctuations. This is detrimental to the sentiment analysis dataset we aim to construct, as we want to cover a wide range of emotional expressions, such as happy, sad, mocking, excited, etc.
> 3. **Missing voice modality**: Some of the videos in Ego-4D lack voices or have muffled voices. Since voice is a crucial modality in sentiment analysis, videos with missing or unclear voices will affect the construction quality of our dataset.
>
> In summary, although the Ego-4D dataset has its advantages in terms of behavioral and scene coverage, it is not suitable for the construction of our specific sentiment analysis dataset due to the lack of rich human interactions, inconspicuous expression of emotions, and incomplete modality of some videos. Therefore, we chose a **self-collected data source** that is more suitable for our research needs.
>
> ### **Q2 How to ensure these videos are eccentric?**
>
> Ensuring that these videos are first-person is a prerequisite for our study. However, to the best of our knowledge, there is currently no particularly well-designed method for detecting whether a video is first-person or not. Here are the specific steps we took to ensure this:
>
> 1. **Selection of data sources**: During the video collection phase, we ensured that most of the collected videos were first-person videos by using **specific keywords** and by selecting **bloggers who frequently post first-person videos** as primary sources. This step played a key role in the initial screening of the dataset.
> 2. **Annotator training**: Before starting the annotation process, we provided detailed training to all annotators by **providing a large number of first-person video examples** so that they could accurately understand and differentiate what a first-person video is. This kind of training helps the annotators to recognize and handle the videos correctly in real life.
> 3. **Strict annotation process**: In the annotation process, for each video, the annotators first **determine whether the video is a first-perspective video and whether the filmmaker is involved in the activities in the video**. If the video does not meet these two criteria, the annotator immediately filters out the video without further processing. This process ensures that only videos that truly meet the requirements are included in the dataset.
> 4. **Sample verification**: To further ensure the quality of the videos and the accuracy of the annotation, we also **performed a sample check on the annotated data**. Through this method, we ensured that the viewpoints of the videos conformed to the definition of the first viewpoint and that the labeling process conformed to the specifications.
>
> Through these measures, we ensured that the viewpoints of the videos in the dataset were correct, and we hope that these explanations have answered your questions.
>
> ### **Q3 Analysis of modality combination**
>
> Thanks for your suggestion, we try the ablation analysis of three modalities: visual, audio and text (subtitles). The experimental results are shown in the following table and Table 4 in rebuttal PDF.
>
> |       Methods       | IC   | DO   | CU   | TUC  | B-3   | B-4   | Rough-L | Cider |
> | :-----------------: | ---- | ---- | ---- | :--: | ----- | ----- | ------- | ----- |
> |    w/o Subtitle     | 2.25 | 2.31 | 2.71 | 2.00 | 20.09 | 15.52 | 36.30   | 81.53 |
> |      w/o Audio      | 2.22 | 2.23 | 2.63 | 2.06 | 18.62 | 14.95 | 34.68   | 81.22 |
> |     w/o Visual      | 1.05 | 1.45 | 1.58 | 1.21 | 9.53  | 6.74  | 29.50   | 31.31 |
> | w/o Subtitle&Audio  | 2.03 | 2.19 | 2.58 | 1.92 | 18.48 | 14.52 | 34.82   | 79.91 |
> | w/o Subtitle&Vision | 0.71 | 0.94 | 1.26 | 0.77 | 9.48  | 6.55  | 28.77   | 30.92 |
> |  w/o Audio&Vision   | 0.96 | 1.45 | 1.44 | 1.18 | 9.50  | 6.75  | 29.33   | 31.12 |
> |        Ours         | 2.39 | 2.39 | 2.77 | 2.18 | 20.64 | 15.99 | 37.20   | 84.06 |
>
> Experimental results show that visual modality is the most important for emotion analysis, and audio modality and text modality of captions can provide more informative context and further improve the performances.

---

> > ### Author Response · Authors · 2024-08-30
> >
> > Dear Reviewer AVha:
> >
> > We are grateful that this paper has overall been well-received by the reviewers. As the end of the discussion period is approaching, we kindly ask you to acknowledge if our rebuttal has addressed your concerns and give us an opportunity to address any further follow-up. Thank you again for your participation.
> >
> > Best
> > Authors

---

> > > ### Comment · Reviewer_AVha · 2024-08-31
> > > **To authors**
> > >
> > > Thank you for your detailed explanation, I will maintain my score.

---

### Official Review · Reviewer_se4o · 2024-07-26
**E3: Exploring Embodied Emotion Through A Large-Scale Egocentric Video Dataset**

**Rating:** 6
**Confidence:** 5
**Correctness:** yes
**Clarity:** yes

**Review:**

•	1.The logical expression of the paper is clear
•	2.The paper proposes a new dataset and publishes four subtasks, which have a strong promoting effect on Understanding human emotions.
•	3.The paper uses an agent to understand human emotions, which has a certain degree of innovation.
Missing references:
SkatingVerse: A large‐scale benchmark for comprehensive evaluation on human action understanding
Anti-uav410: A thermal infrared benchmark and customized scheme for tracking drones in the wild
Anti-uav: a large-scale benchmark for vision-based uav tracking

**Strengths:**

•	The article has a strong contribution to the dataset.
•	The use of agents for emotion recognition tasks is relatively novel.

**Additional Feedback:**

•	1.More experiments should be added.
•	2.The correctness of the URL Link for the project requires further confirmation.

**Documentation:**

•	No. There seems to be an issue with the URL.

**Limitations:**

•	Whether using large model to do video task need to consider the problem of reasoning time, whether it has application value in actual deployment.

**Opportunities For Improvement:**

•	The experiments in the article can be conducted more extensively
•	You can add some visual experimental results

**Relation To Prior Work:**

yes

**Summary And Contributions:**

Briefly summarize the submission and its contributions. If desired, add formatting using Markdown and formulas using LaTeX. For more information see https://openreview.net/faq
1.	A large-scale collection of FPV videos that capture people’s activities and emotions in daily life.
2.	The introduction of four benchmark tasks: emotion recognition,emotion classification, emotion localization, emotion reasoning.
3.	A video understanding framework, named Emotion LlaMa.

---

> ### Author Rebuttal · Authors · 2024-08-17
>
> # **For Reviewer se4o**
>
> Thank you for recognizing our paper and recommending it for acceptance. Now, we will address the key arguments raised in the reviews.
>
> ### **Q1 More extensively experiments**
>
> Thanks to your suggestion, we have made every effort to include 4 additional experiments in the rebuttal PDF, including:
>
>  1) **two comparison experiments** on the common emotion analysis dataset (**see Table1 & 2, in rebuttal PDF**).
>
> 2) **an audio ablation experiment** on embodied emotion classification tasks (**see Table3, in rebuttal PDF**).
>
> 3) **an additional ablation experiment** on modal combination analysis (**see Table4, in rebuttal PDF**).
>
> The additional ablation and comparison further demonstrate the effectiveness of the Emotion-LlaMA framework.
>
> ### **Q2 Adding more references**
>
> Thanks for reminding us of the relevant work we missed, we'll cite the relevant articles in the final version
>
> ### **Q3 Some visual experimental results**
>
> Due to space constraints, we have included the visualizations **in Appendix E, Figs. 4-7.** We have additionally included a set of visualizations **in the rebuttal PDF, see Fig. 1.** In the final version, we will adjust the structure of the paper accordingly to incorporate some of the visualizations into the main paper.
>
> ### **Q4 The correctness of the URL Link**
>
> Thank you for your careful review, we have double-checked that all the URLs are opening correctly. Could you kindly remind us which URL has the problem? We will fix it immediately.
>
> ### **Q5 About the reasoning time**
>
> While significant progress has been made in multimodal large language models nowadays, video large language models are remaining to be explored. One of the limitations is the more massive information of video compared to images, and existing works such as MiniGPT4-Video [1], [2] are devoted to **reducing the number of video tokens** to speed up the processing of video tasks. For embodied sentiment analysis, it is a challenge to reduce the inference time, which we leave for future work.

---

> > ### Author Response · Authors · 2024-08-30
> >
> > Dear Reviewer se4o:
> >
> > We are grateful that this paper has overall been well-received by the reviewers. As the end of the discussion period is approaching, we kindly ask you to acknowledge if our rebuttal has addressed your concerns and give us an opportunity to address any further follow-up. Thank you again for your participation.
> >
> > Best
> > Authors

---

### Author Rebuttal · Authors · 2024-08-17

### **For ALL Reviews**



We extend our heartfelt gratitude to ALL reviewers for their insightful comments and unanimously acknowledgment of our work's:



**[ALL Reviews]** **Clear and meaningful dataset contribution:** We are grateful for the reviewers’ positive feedback on our impressive experimental results, particularly our method's ability to consistently outperform existing approaches in both objective metrics and user studies.



**[ALL Reviews] Notable for the modalities and data diversity**: We are delighted that most reviewers acknowledge our motivation is good and interesting, and the novelty and effectiveness of our method in action image generation.



**[ALL Reviews]** **Detailed and Accessible Presentation:** We appreciate the reviewers highlighting the comprehensiveness of our explanations and the ease of understanding our paper.



We will now address the key points raised in the reviews and provide detailed responses to each reviewer. We highly recommend that reviewers take the time to visit our PDF for additional visualization of results.

Once again, we extend our thanks for the reviewers’ time and valuable insights, and we look forward to any additional feedback or questions regarding our work.

---

### Decision · Program_Chairs · 2024-09-26

**Decision:**

Accept (Poster)

**Comment:**

The authors proposed the first large-scale first-person view (FPV) video dataset, consisting visual, acoustic, and textual modalities, designed to explore embodied emotion. All the reviewers acknowledge  that relying on agents to interact naturally for emotion recognition is a fresh approach. One baseline namely Emotion-LlaMa integrates visual, acoustic, and textual modalities for emotion related task. The crowdsausing annotation method may not be accurate, especially for the emotion tasks. Also the emotion only classified into negative and positive, which is coarse. Finer-grained expression of emotion should be considered.

Considering the reviewers’ comments, I think that the paper does have merit. As such, I am recommending acceptance.